# ZNF143 mediates CTCF-bound promoter–enhancer loops required for murine hematopoietic stem and progenitor cell function

Qiling Zhou [1,2], Miao Yu[3], Roberto Tirado-Magallanes[1], Bin Li[4], Lingshi Kong[1], Mingrui Guo[1,2], Zi Hui Tan[1], Sanghoon Lee[1,12], Li Chai[5], Akihiko Numata[1,6], Touati Benoukraf [1,7], Melissa Jane Fullwood[1,8,9], Motomi Osato [1], Bing Ren [4,10] & Daniel G. Tenen [1,11✉]

CCCTC binding factor (CTCF) is an important factor in the maintenance of chromatin–chromatin interactions, yet the mechanism regulating its binding to chromatin is unknown. We demonstrate that zinc finger protein 143 (ZNF143) is a key regulator for CTCF-bound promoter–enhancer loops. In the murine genome, a large percentage of CTCF and ZNF143 DNA binding motifs are distributed 37 bp apart in the convergent orientation. Furthermore, deletion of ZNF143 leads to loss of CTCF binding on promoter and enhancer regions associated with gene expression changes. CTCF-bound promoter–enhancer loops are also disrupted after excision of ZNF143. ZNF143-CTCF-bound promoter–enhancer loops regulate gene expression patterns essential for maintenance of murine hematopoietic stem and progenitor cell integrity. Our data suggest a common feature of gene regulation is that ZNF143 is a critical factor for CTCF-bound promoter–enhancer loops.

[1] Cancer Science Institute of Singapore, National University of Singapore, 117599 Singapore, Singapore. [2] YLL School of Medicine, National University of Singapore, 119228 Singapore, Singapore. [3] School of Life Sciences, Fudan University, Shanghai, China. [4] Ludwig Institute for Cancer Research, La Jolla, CA 92093, USA. [5] Department of Pathology, Brigham and Women's Hospital, Boston, MA 02115, USA. [6] Medicine and Biosystemic Science, Kyushu University Graduate School of Medical Science, Fukuoka, Japan. [7] Division of BioMedical Sciences, Faculty of Medicine, Memorial University of Newfoundland, St. John's, NL A1B 3V6, Canada. [8] School of Biological Sciences, Nanyang Technological University, 60 Nanyang Drive, 637551 Singapore, Singapore. [9] Institute of Molecular and Cell Biology, Agency for Science, Technology and Research (A*STAR), 61 Biopolis Drive, Proteos 138673, Singapore. [10] Department of Cellular & Molecular Medicine, Moores Cancer Center and Institute of Genome Medicine, UCSD School of Medicine, 9500 Gilman Drive, La Jolla, CA 92093, USA. [11] Harvard Stem Cell Institute, Harvard Medical School, Boston, MA 02115, USA. [12] Present address: Department of Systems Biology, The University of Texas MD Anderson Cancer Center, Houston, TX 77030, USA. ✉email: daniel.tenen@nus.edu.sg

Gene transcription is commonly regulated by chromatin loops between promoters and regulatory elements such as enhancers and silencers[1–3]. Different transcription factors (TFs) bind to these regulatory elements to regulate promoter–enhancer loops in a cell-type-specific manner. Although they play an essential role in control of gene expression, how TFs contribute to promoter–enhancer loops is not well understood.

In addition to its essential role in maintaining Topologically Associating Domains (TADs)[4–9], several lines of evidence suggest that CCCTC binding factor (CTCF) plays a critical role in promoter–enhancer loop regulation as well. First, down-regulation of CTCF by either targeted degradation or RNAi decreases the intensity of enhancer–promoter loops, consistent with high enrichment of CTCF motifs within promoter regions[4,5,7–10]. Second, detection of CTCF in protein complexes enriched on active promoters by ChIP-MS supports that CTCF is involved in enhancer–promoter loops[11].

While it has been well-established that CTCF mediates DNA–DNA interactions by forming dimers that directly bind to both sides of the DNA[12,13], the regulatory mechanism controlling CTCF–DNA binding remains unknown. One hypothesis is that other factors such as cohesin play roles to mediate CTCF–DNA binding. Although multiple studies indicate that CTCF and cohesin interact with each other to contribute to chromatin structure maintenance[6,14,15], a recent crystallographic structural study demonstrated that a specific CTCF mutation abolished CTCF–cohesin interaction, and further affected cohesin localization on CTCF sites, whereas CTCF binding on its sites remained unchanged[16]. This study indicates that CTCF recruits cohesin to its binding site, but CTCF–cohesin interaction does not regulate CTCF–DNA binding. So, to date it is not known which factor(s), if any, mediate CTCF–DNA binding to regulate chromatin interactions and gene expression.

The zinc finger protein 143 (ZNF143/ZFP143), was first discovered as selenocysteine tRNA gene transcription activating factor (STAF) in *Xenopus laevis* functioning as a sequence specific transcriptional activator of both Pol II and Pol III[17,18]. Later on, studies reveal that as a co-activator, ZNF143 might be a key regulator in T lymphoblastic leukemia (TLL) through its association with Notch1[19,20]. Despite bioinformatic analysis suggesting that, similar to cohesin, ZNF143 co-localizes with CTCF[7,21,22], and knocking down ZNF143 affects chromatin interaction at individual loci[22], how CTCF and ZNF143 regulate the function of each other remains unknown. Building upon our previous work identify ZNF143 as an important regulator of *Cebpa* expression[23], in this study, we used a hematopoietic model system to investigate whether ZNF143 mediates gene transcription through CTCF in a general manner.

## Results

### CTCF genome-wide DNA binding is dependent on ZNF143.
Since ZNF143 and one of its family members, zinc finger protein 76 (ZNF76), share over 60% protein sequence identity and bind in vitro to the same DNA sequence[17,23], whereas most commercial anti-ZNF143 antibodies recognize both ZNF143 and ZNF76 (data not shown), we first generated a specific anti-ZNF143 rabbit polyclonal antibody which only recognizes ZNF143, and not ZNF76, in both Western-blot and IP (Supplementary Fig. 1A, B). Then, we generated conditional *Znf143* knockout mice to cross with *Rosa26ERT2-cre* mice so that *Znf143* deletion can be induced by tamoxifen injection. After systematic time and dosage testing, a successful excision of *Znf143* in adult hematopoietic stem and progenitor cells (HSPCs) was confirmed at the level of both RNA and protein (Supplementary Fig. 1C–G).

Under such conditions, we performed both ZNF143 and CTCF ChIP-seq in HSPCs. Surprisingly, we observed that, after *Znf143* depletion, dramatic loss of the CTCF ChIP-seq signal was detected in regions where both ZNF143 and CTCF were enriched in wild type cells, without affecting CTCF mRNA or protein levels (Fig. 1a, Supplementary Fig. 2A, B; Supplementary Data 1). In contrast, no significant changes in CTCF localization were detected in regions where CTCF bound alone (Fig. 1a, Supplementary Data 1). Little is known regarding the regulation of CTCF loading onto chromatin by other factors, even though it is reported that ZNF143 and CTCF are both chromatin structure factors[7,21,22]. One possibility is that by forming a complex with CTCF, ZNF143 stabilizes the binding of CTCF on their co-binding regions. Indeed, reciprocal co-IP confirmed that ZNF143 and CTCF interacted with each other endogenously by immunoprecipitating either ZNF143 or CTCF (Fig. 1b). Furthermore, motif analysis using ZNF143 and CTCF ChIP-seq results from HSPCs found that ZNF143 and CTCF motifs were highly enriched within a 100 bp distance in the mouse genome (Supplementary Fig. 2C, Supplementary Data 2). In contrast, an equal distribution within ±2 kb regions was demonstrated with mutant ZNF143 motifs, which were randomly generated by Regulatory Sequence Analysis Tools (RSAT), a suite of bioinformatics tools to detect and analyze of *cis*-regulatory elements in genome sequences[24] (Supplementary Fig. 2C). Surprisingly, deep analysis revealed that among 17,319 ZNF143 motifs located within 100 bp from CTCF motifs, 10,544 ZNF143 motifs (60.88%) were located 37 bp away from the nearest CTCF motifs (Fig. 1c, Supplementary Data 2). There is precedent for a TF being stabilized on its bound DNA by another factor bound nearby[25], therefore the short distance between ZNF143 and CTCF binding motifs provided a hint about how ZNF143 regulated CTCF binding on their co-binding regions. Since it is known that the orientation of CTCF motifs is a key factor for chromatin loop formation[7,8,26,27], we were wondering whether ZNF143 and CTCF motifs were oriented in a pattern. Four possible patterns exist between these two motifs as they are not palindromic: ZNF143 and CTCF motifs are 1) convergent on opposite strands (convergent); 2) divergent on opposite strands (divergent); 3) both on forward strand (F-F); 4) both on reverse strand (R-R). After analyzing 10,544 pairs of ZNF143 and CTCF motifs located 37 bp away from each other, we observed 99.9% (10,529) formed the convergent orientation (Fig. 1c, Supplementary Fig. 2C). With such a remarkably consistent pattern between ZNF143 and CTCF motifs across genome, we hypothesized that ZNF143 and CTCF work together genome-wide for loop formation and gene expression. On the other hand, we also queried whether CTCF facilitates ZNF143 binding. We observed higher enrichment of ZNF143 on ZNF143-CTCF shared sites compared to ZNF143-alone sites, suggesting that ZNF143 and CTCF regulate the binding of one another (Supplementary Fig. 2E). To further investigate which genomic regions ZNF143 and CTCF co-bound, ChIP-seq of ZNF143 and CTCF were further analyzed and a strong global correlation was detected, consistent with previous bioinformatic studies (Supplementary Fig. 2F–I)[21,22]. Since approximately 90% of ZNF143 and CTCF co-bound regions were on promoter/enhancer regions (Supplementary Figs. 2D, G), we wondered whether ZNF143 is involved in CTCF-mediated transcriptional regulation. Active promoter scanning analysis confirmed that both ZNF143 and CTCF were highly enriched on active promoter regions compared to inactive promoter regions (Fig. 1d), and that ZNF143 was significantly enriched on CTCF-bound enhancer regions (Fig. 1e), indicating the co-localization of ZNF143 and CTCF on both promoter and enhancer regions.

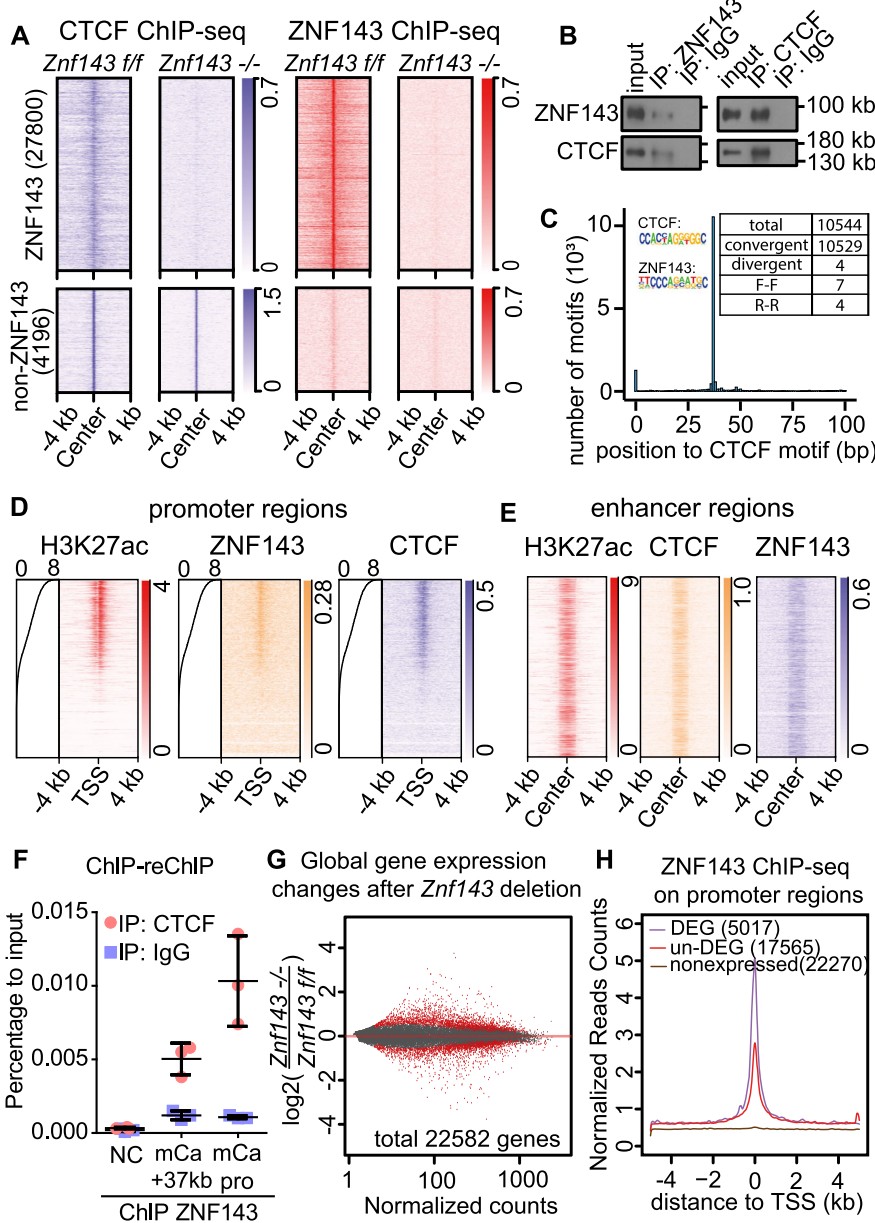

**Fig. 1 ZNF143 stabilizes CTCF binding on promoter and enhancer regions to regulate gene transcription. a** Heatmaps of ZNF143 and CTCF ChIP-seq in ZNF143 wild type (*Znf143 f/f*) and ZNF143 knockout (*Znf143 −/−*) murine HSPCs. **b** Reciprocal co-IP of ZNF143 and CTCF in murine HSPCs. Experiment is repeated independently for three times with similar results. **c** Histogram describing the distance between CTCF and ZNF143 motifs on the murine genome (mm10). The *X*-axis represents the relative distance between ZNF143 and CTCF motif, while the *Y*-axis represents the number of ZNF143 motifs in each bin. Each bin size is 1 bp. The table presents the number of ZNF143 motifs in the different orientation patterns to the nearest CTCF motifs within 10,544 ZNF143 motifs located 37 bp apart from the nearest CTCF motifs. **d** Heatmap of H3K27ac, ZNF143, and CTCF ChIP-seq in *Znf143 f/f* murine HSPCs within the regions ±4 kb from the transcription start site (TSS). Gene order is ranked by RNA expression from high to low in *Znf143 f/f* cells (heatmap from top to bottom). **e** Heatmap of H3K27ac, ZNF143, and CTCF ChIP-seq in *Znf143 f/f* murine HSPCs within ±4 kb from CTCF-bound enhancer regions. **f** ChIP-reChIP of ZNF143 and CTCF on a negative intergenic control region (NC), *Cebpa* promoter region (mCa pro), and *Cebpa* +37 kb enhancer region (mCa + 37 kb) in *Znf143 f/f* murine HSPCs. *Y*-axis: the mean ± SD from triplicate experiments. **g** MA plot demonstrating gene expression changes between *Znf143 f/f* and ZNF143 −/− murine HSCs. Red dots represent differentially expressed genes (*p* < 0.05). *n* = three biological replicates. *P*-value is determined by the Wald test. **h** The mean plot of ZNF143 ChIP-seq within ±4 kb TSS regions in *Znf143 f/f* murine HSPCs. Genes are divided into (1) differentially expressed genes (DEG), (2) non-differentially expressed genes (un-DEG), and 3) non-expressed genes (non-expressed) as defined in (**g**). HSPCs: c-kit+, lineage- cells; HSCs: LSK+ (lineage-, Sca-1+, c-kit+), CD150+, CD48- cells. ChIP-seq in panels (**a**), (**c**), (**d**), (**e**), and (**h**) was performed in two biological replicates. Result was shown in one replicate.

To address the limitation of ChIP-seq, whose resolution is modest and therefore apparent co-localization may not necessarily indicate close association or interaction, we performed ChIP–reChIP for further validation. Consistent with our hypothesis, enrichment on promoter and enhancer regions were detected in CTCF pull-down but not IgG control in the elute from ZNF143 ChIP (Fig. 1f). Together with our co-IP results (Fig. 1b), it demonstrated that ZNF143 and CTCF form a complex to bind to both promoter and enhancer regions. Furthermore, over 5,000 genes displayed significant expression

changes after *Znf143* deletion, corresponding with high ZNF143 ChIP-seq signals detected on their promoter regions (Fig. 1g, h, Supplementary Data 3). Taken together, our results indicate that ZNF143 is a critical mediator for CTCF-involved gene transcription by regulating CTCF binding on promoter and enhancer regions.

**ZNF143 depletion affects CTCF-bound promoter–enhancer loop intensities while compartments and TADs remain unchanged.** Next, we investigated whether CTCF-bound chromatin-chromatin interactions were disrupted after ZNF143 depletion by performing in situ HiC using *Znf143 f/f* and *Znf143 −/−* HSPCs. Within the 11,930 loops detected in *Znf143 f/f* cells at 5 kb resolution, 4017 loops significantly decreased in *Znf143 −/−* cells. Further annotation of these affected loops revealed that 38% of them mediated promoter–enhancer and 10.7% mediate promoter–promoter interaction, suggesting that the loss of ZNF143 mainly affected promoter–enhancer loops (Fig. 2a, b, Supplementary Data 4, 5). One alternative possibility is that the loss of ZNF143 affects the expression of some other chromatin structure factor, leading to indirect changes of promoter–enhancer loops. To exclude this possibility, after dividing loops into (1) differentially changed and (2) unchanged groups, CTCF motif scanning was performed on their anchors, followed by ZNF143 ChIP-seq analysis on regions ± 2 kb from these motifs (Supplementary Data 6). The results demonstrated that consistent with loop intensity changing, ZNF143 was highly enriched around these motif regions located on differential loop anchors, compared to anchors of unchanged loops (Fig. 2c, d). In line with these findings, we also detected significantly decreased CTCF binding on ZNF143-dependent loop anchors, suggesting that ZNF143 directly participated in the maintenance of CTCF-involved promoter–enhancer loops (Fig. 2e, Supplementary Fig. 3A, B).

Since our previous study revealed that ZNF143 is important for *Cebpa* regulation[23], and that ZNF143 and CTCF were detected on both the *Cebpa* promoter and a previously reported +37 kb enhancer region (DRE 37 kb region) (Fig. 1f), the *Cebpa* locus was selected for further validation. By performing chromosome conformation capture (3 C), we confirmed the decrease of interaction frequency between the *Cebpa* promoter and the 37 kb *Cebpa* enhancer after deletion of ZNF143 (Fig. 2f). In agreement with the loop change, significant down-regulation of *Cebpa* mRNA levels and decreases of ZNF143 and CTCF enrichment on both the *Cebpa* promoter and enhancer were detected (Fig. 2g, h), indicating that loss of ZNF143 affected CTCF binding on both the *Cebpa* promoter and enhancer, leading to down-regulation of *Cebpa* expression by decreasing loop frequency. Taken together with our other results, we conclude that ZNF143 mediates CTCF binding on chromatin to regulate promoter–enhancer loops, thus mediating gene transcription.

Since the majority of ZNF143 binding occurred in promoter regions (Supplementary Fig. 2D, F, G), we asked whether ZNF143 was only involved in CTCF-bound promoter–enhancer loops, but not large-scale chromatin structures such as compartmentalization and TAD formation. Indeed, we did not observe significant changes in compartmentalization (Fig. 3a, Supplementary Fig. 4A). Moreover, genome-wide contact-probability scanning revealed no difference between *Znf143 f/f* and *Znf143 −/−* HSPCs (Fig. 3b). We conclude that loss of ZNF143 does not affect the global distribution of active and inactive chromatin. Next, we zoomed in to analyze TAD formation. TADs are detected with a pair of strong boundaries, or insulation scores, and typically span a distance from approximately 100 kb to 1 Mb in murine and human genomes. They are characterized by the feature that two

loci within the same TAD interact with each other more frequently than two loci in different TADs[4,5]. Using the bioinformatic tool Arrowhead to define TADs[7], we detected 3,086 TADs in *Znf143 f/f* HSPCs and 3,114 TADs in *Znf143 −/−* (Supplementary Fig. 4B, Supplementary Data 7), indicating that loss of ZNF143 did not affect TAD numbers. Furthermore, no significant changes in TADs were detected globally (Fig. 3c, Supplementary Fig. 4C). Insulation score analysis was also performed to double confirm no change in TADs formation (Fig. 3d)[28]. In line with that observation, 2,331 CTCF-alone peaks and less than 400 CTCF-ZNF143 shared peaks were detected on TAD boundaries (Supplementary Fig. 4D), indicating that ZNF143 is a novel factor mediating CTCF-bound loops but not TADs.

**ZNF143 is essential for hematopoietic stem and progenitor cells integrity.** Considering the genome-wide regulation of CTCF-bound promoter–enhancer loops and gene expression, we were interested in assessing the biological effects of loss of *Znf143*, using the hematopoietic system as a model. As we observed a reduction in HSPC populations at a late time point after the deletion of *Znf143* (Supplementary Fig. 1D), we next performed systematic analysis to examine the biological effects of loss of *Znf143* in hematopoiesis. The *Vav1-iCre* mouse model is widely used for studies of hematopoiesis in the embryonic stage as it starts expressing at E9.5 and is greatly enhanced at E12.5 in hematopoietic cells[29]. We crossed *Znf143 flox/flox* mice with *Vav1-iCre* mice to investigate the phenotype in hematopoiesis during embryonic development. In line with much paler embryos at E14.5, smaller and paler *Znf143 −/−* fetal livers were observed, consistent with a significant decrease in cell counts from whole fetal livers (Fig. 4a, b). Furthermore, no *Znf143 −/−* mice were found among 65 viable progenies (Supplementary Fig. 5A), indicating that loss of *Znf143* induced a severe embryonic anemia and lethality. Next, the role of ZNF143 in adult hematopoiesis was examined by crossing *Znf143 flox/flox* mice with *Mx1-Cre* mice for inducible deletion through poly I:C injection, and reduction of ZNF143 protein was detected in HSPCs 8 days after the first poly I:C injection (Supplementary Fig. 1G). After inducible deletion of *Znf143*, detection of a significant reduction in white blood cell counts indicated its critical role in adult hematopoiesis (Supplementary Fig. 5B). To systematically investigate which subpopulations were affected by the loss of ZNF143, multiple stem cell and progenitor cell populations were profiled after *Znf143* excision using both fetal liver cells and total bone marrow cells (BMCs). Consistent results obtained using both embryonic and adult hematopoietic cells demonstrated that long-term hematopoietic stem cells (LT-HSCs), short-term hematopoietic stem cells (ST-HSCs), multipotent progenitor cells (MPPs), common myeloid progenitor cells (CMPs), granulocyte progenitor cells (GMPs), and megakaryocyte-erythroid progenitor cells (MEPs) were all dramatically reduced after *Znf143* excision (Fig. 4c, d, Supplementary Fig. 5C, D), suggesting the key role of ZNF143 in the maintenance of the integrity of HSPCs. To exclude the potential hematopoietic cell-extrinsic effects of loss of *Znf143*, competitive transplantation was performed using adult bone marrow cells from *Znf143 f/f, Mx1-Cre +* or *Znf143 f/f* mice. After detection of successful engraftment, *Znf143* excision was induced by poly I:C injection, followed by analysis of peripheral blood (PB) chimerism over 4 months. In agreement with the results obtained with fetal liver cell transplantation (Supplementary Figs. 5E, F, 6), continuous reduction of engraftment was observed after deletion of *Znf143*, whereas wild type cells succeeded in reconstituting the hematopoietic system over 4 months (Fig. 4e, f).

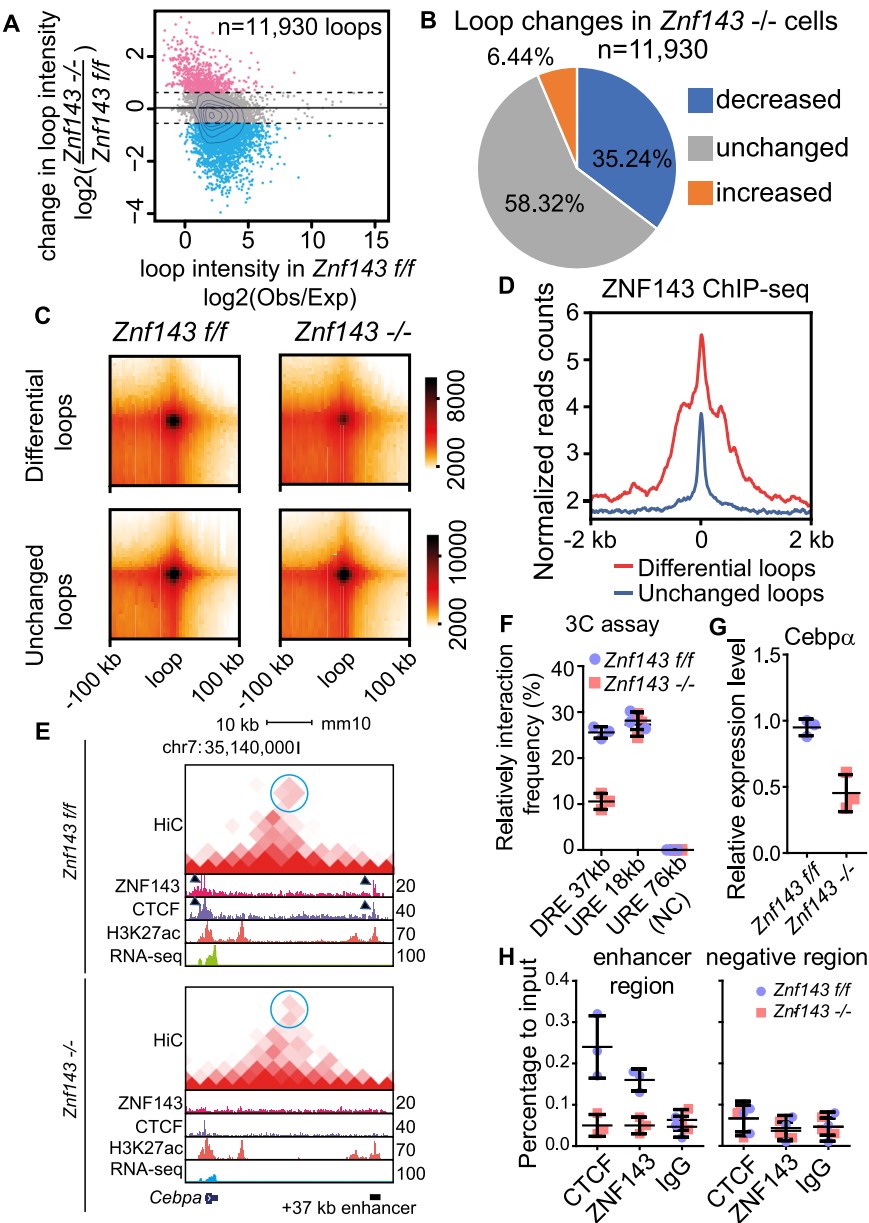

**Fig. 2 Promoter–enhancer loops are maintained by ZNF143. a** Density plot describes individual intensity change of loops detected in *Znf143 f/f* murine HSPCs (LSK+) after deletion of *Znf143*. Red dots present loops with increased intensity (1.5-fold threshold, 768 loops), while blue dots present loops with decreased intensity (-1.5-fold threshold, 4204 loops). **b** Pie chart describes a statistical summary of loop changes using the same categories described in (**a**). **c** Aggregate peak analysis for intensity of differential loops. Loops divided into different groups as described in (**a**) are aggregated at the center of a 100 kb window in 5 kb resolution. **d** Mean plot of ZNF143 ChIP-seq within ±2 kb from CTCF motifs on loop anchors. Loops are divided into different groups as described in (**a**). *n* = two biological replicates. **e** UCSC genome browser shot presents HiC interaction frequencies, ChIP-seq profiles of ZNF143, CTCF, and H3K27ac, and RNA expression on *Cebpa* locus in both *Znf143 f/f* and *Znf143 −/−* murine HSPCs. The balanced HiC two-dimensional contact matrix is plotted on top. The color intensity presents interaction frequency, while blue circles indicate the *Cebpa* 37 kb enhancer–promoter loop. ZNF143 and CTCF binding profile are presented below, while H3K27ac histone mark indicates an active chromatin status. Dark triangles indicate ZNF143 and CTCF peaks located on loop anchors. The maximum *Y*-axis value of ChIP-seq signal is set as indicated. **f** The interaction frequencies between the *Cebpa* promoter and the *Cebpa* +37 kb enhancer ("DRE 37 kb") or with a non-related control region located 18 kb upstream from the *Cebpa* promoter ("URE 18 kb") were examined by 3 C qPCR in both *Znf143 f/f* and *Znf143 −/−* murine HSPCs. The 76 kb upstream region from the *Cebpa* promoter (URE 76 kb) is presented as a negative control region (NC). **g** Dot plot describes relative *Cebpa* mRNA levels in *Znf143 f/f* and *Znf143 −/−* murine HSCs (LSK+, CD150+, CD48−) normalized to *β-actin*. **h** ChIP-qPCR of ZNF143 and CTCF on the *Cebpa* +37 kb enhancer region or intergenic negative control region. All dot plots are presented in mean ± SD from three biological replicates experiment.

These results indicated that ZNF143 is a key cell-intrinsic factor maintaining HSPC integrity.

**ZNF143-CTCF loops regulate pathways critical for hematopoiesis.** Finally, we sought to identify the genes and pathways

regulated by ZNF143-CTCF loops contributing to the murine phenotype. By integrating gene expression and ZNF143 localization on promoter regions, we narrowed the number of high-confidence ZNF143 direct target genes down to 1467 (Supplementary Data 8). We observed higher global expression levels

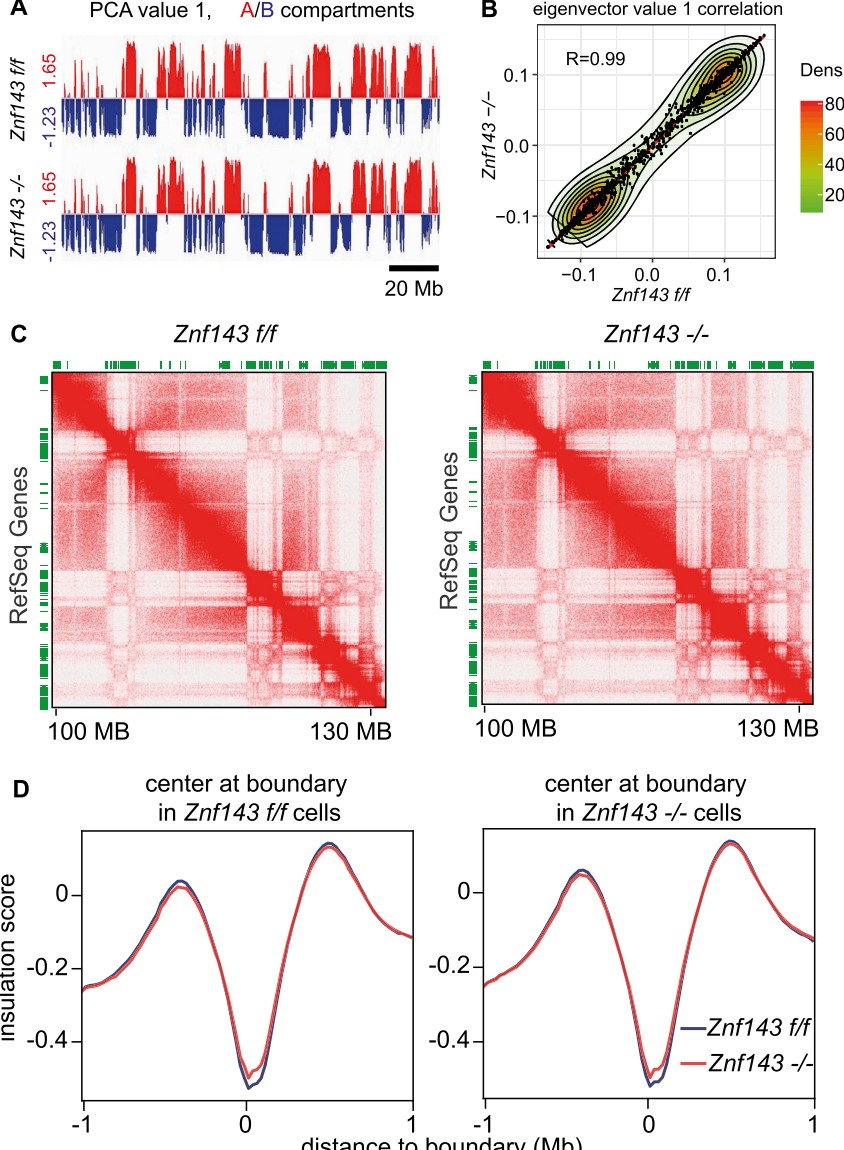

**Fig. 3 Chromosome compartments and TADs mainly remain unchanged after Znf143 deletion. a** PCA analysis for the first principle component representing compartment A and B on chr1 at 50 kb resolution. **b** Density plot describes global correlation between the eigenvector value from wild type (*Znf143 f/f*) and *Znf143* depleted (*Znf143 −/−*) LSKs at 1 Mb resolution. The *X*-axis represents the eigenvector value in *Znf143 f/f* cells on each genome locus, while the *Y*-axis the value in *Znf143 −/−* cells in the same locus. **c** Comparison of TADs in wild type (*Znf143 f/f*) and *Znf143* knockout (*Znf143 −/−*) cells. A snapshot of balanced contact matrices on chr1:100–130 Mb. **d** Mean plot describes genome-wide insulation score around TADs boundaries. The *X*-axis represents the genome distance to the TADs boundaries, while the *Y*-axis the insulation score. The 0 position on the *X*-axis in the left panel indicates TAD boundaries in *Znf143 f/f* cells, while in the right panel TAD boundaries in *Znf143 −/−* cells.

among these ZNF143 direct targets compared to all other genes, and these ZNF143 targets were also significantly downregulated after deletion of *Znf143* (Fig. 5a). Principal component analysis (PCA) indicated different clustering between wild type and *Znf143* deleted cells (Supplementary Fig. 7A, B). Furthermore, multiple pathways (such as ribosomal pathways, metabolism-related pathways, and hematopoietic stem cell-related pathways) and genes critical for hematopoiesis (RUNX1 and Gata2, among others) were downregulated after the loss of *Znf143* (Fig. 5b, Supplementary Fig. 7C–E). These genes were reported as key regulators of hematopoietic stem cells, and mutations of these genes and disorders of these pathways are commonly observed in bone marrow failure, a disease syndrome defined as

a drastic decline in the ability of bone marrow to produce mature blood[30,31]. Consistent with the observation of dramatic disappearance of mature blood cells after excision of *Znf143* (Fig. 4a, b, Supplementary Fig. 5B), our data suggested that ZNF143 maintained HSPCs integrity by regulating key genes and pathways. Furthermore, after integrating global loops and CTCF binding, we confirmed a significant reduction of CTCF binding on the promoter–enhancer loop anchors on ZNF143 regulated genes, consistent with decreased loop intensities (Fig. 5c, d, Supplementary Fig. 7F). Together, these results indicate that ZNF143-CTCF-mediated promoter–enhancer loops play important roles to regulate key genes and pathways essential for the maintenance of HSPC integrity.

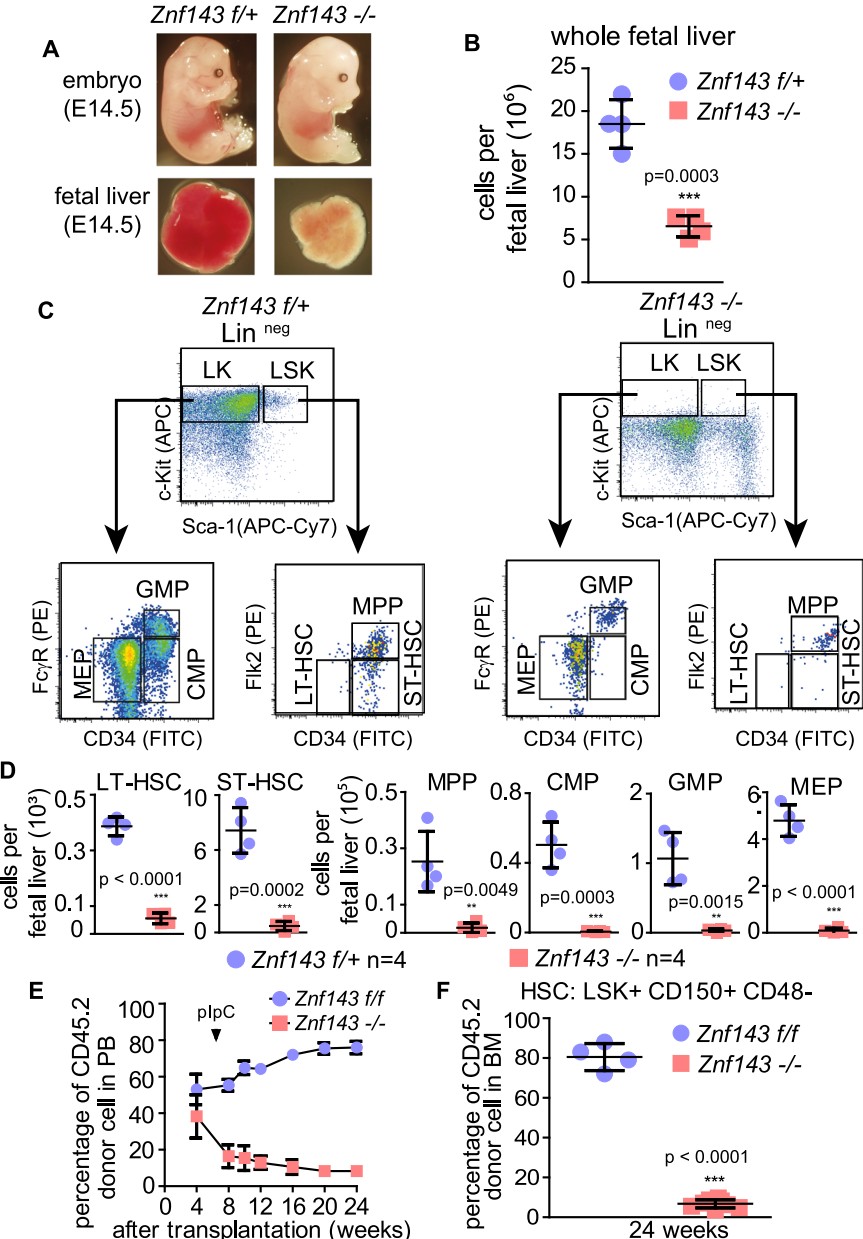

**Fig. 4 ZNF143 is essential for maintenance of HSPC integrity. a** Image of embryos and fetal livers of *Znf143 f/+* and *Znf143* conditional knockout (*Znf143 −/−*) mice at embryonic day 14.5 (E-14.5). The floxed *Znf143* allele was deleted specifically in hematopoietic cells by crossing with *Vav1-iCre* mice. **b** Dot plot describes absolute numbers of nucleated cells in *Znf143 f/+* and *Znf143 −/−* fetal liver at E-14.5 (*n* = 4 in each group). Values are presented as mean ± SD. **c** Dot plot of flow cytometry analysis of different HSPC populations in *Znf143 f/+* and *Znf143 −/−* fetal liver cells at E-14.5. The upper left panel demonstrates the LSK cells (Lin- c-Kit+ Sca-1+) or LK cells (Lin- c-Kit+) which were further analyzed in the lower-left panel. **d** Absolute numbers of HSPCs as defined in (**c**), including LT-HSC (LSK CD34− Flk2−), ST-HSC (LSK CD34+Flk2−), MPP (LSK CD34+Flk2+), CMP (LK CD34+FcγR−), GMP (LK CD34+FcγR+), and MEP (LK CD34− FcγR−), comparing fetal liver cells from *Znf143 f/+* with *Znf143 −/−* embryos. Data were obtained from four pairs of littermates. Values are presented as mean ± SD. **e** Donor chimerism in peripheral blood (PB) for 24 weeks after transplantation with *Znf143 f/f* or *Znf143 f/f, Mx1-Cre+* whole bone marrow cells after removal of red blood cells. *Znf143* was deleted by pIpC injection for three consecutive days 2 weeks after transplantation. Donor chimerism was monitored by measuring the percentage of CD45.2 positive cells in the peripheral blood at the indicated time points. *Znf143 f/f*: *n* = 4, *Znf143 −/−*: *n* = 8. Values are presented as mean ± SD. **f** Dot plot demonstrating the donor chimerism of HSCs (CD150+CD48− LSK+) in bone marrow (BM) 24 weeks after transplantation. Each dot represents the percentage of CD45.2 positive HSCs in one recipient mouse. *Znf143 f/f*: *n* = 4, *Znf143 −/−*: *n* = 8. Values are presented as mean ± SD. *P* values in (**b**), (**d**), and (**f**) are determined by the two tailed unpaired *t* test (*$p < 0.05$, **$p < 0.01$, ***$p < 0.001$).

## Discussion

As a key factor in the maintenance of chromatin structure, ubiquitously expressed in most organs and highly conserved in higher eukaryotes, CTCF is critical for multiple cellular processes. In addition to resulting in disruption of blastocyst stage processing and exhibiting early embryonic lethality, deletion of CTCF also results in defects of lineage differentiation[32,33]. Although it is widely reported that CTCF and other factors such as cohesin co-localize together to mediate chromatin-chromatin interactions, a recent study of the crystallographic structure of the

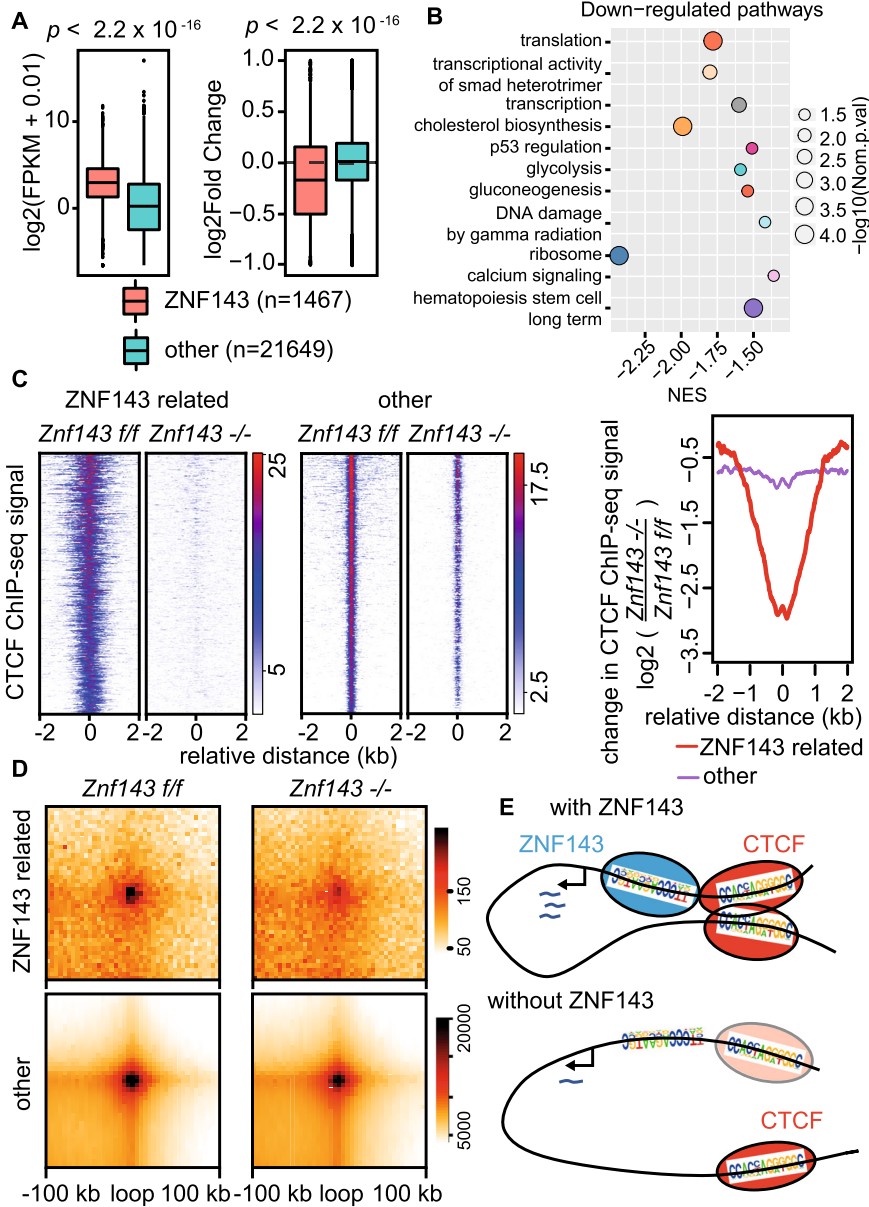

**Fig. 5 ZNF143-CTCF-bound loops regulate key genes and pathways contributing to the stem cell phenotype. a** Left panel, box plot describes log2 fragment per kb per million (FPKM) of genes *in Znf143 f/f* HSCs that are 1) highly enriched of ZNF143 on their promoters (ZNF143, 1467 genes) or 2) not associated with high ZNF143 enrichment (other, 21,649 genes). Right panel, log2 fold change in expression of genes grouped as in left panel. *P* values determined by the two-sided Mann Whitney Wilcoxon Test. *n* = three biological replicates. Boxplot represents median and the 25th and 75th percentiles of the distribution. The upper and lower whisker represents data no further than 1.5 times the IQR from 25th or 75th percentiles. **b** Bubble plot describes downregulated pathways after *Znf143* excision discovered by Gene Set Enrichment Analysis (GSEA). The *X*-axis represents normalized enrichment score (NES), while bubble size represents normalized *p*-value (transformed into -log10 format). *P*-value is determined as described previously[52]. **c** Left panel, heatmap of the CTCF ChIP-seq signal within ±2 kb from CTCF motifs on loop anchors. Each column represents the relative distance to the CTCF motif, while each row represents normalized read counts of CTCF on the loop anchor region. Loops are divided into two groups according to ZNF143 binding on their anchors. The right panel demonstrates the mean plot of the average changes of the CTCF ChIP-seq signal within ±2 kb from CTCF motifs on loop anchors. *n* = two biological replicates. **d** Aggregate peak analysis demonstrates the intensity change of loops grouped as described in (**c**). Loop intensities are aggregated at the center of a 100 kb window in 5 kb resolution. **e** Proposed model depicting regulation of CTCF-bound loops by ZNF143.

CTCF–cohesin interaction demonstrated that this interaction does not regulate CTCF–DNA binding[4–8,14,16].

A previous study suggested that zinc finger TF ZNF143 is involved in promoter–enhancer loops[22]; however, how it regulates such loops is unknown. In this study, we further investigate the function between ZNF143 and CTCF and demonstrate that ZNF143 is a key regulator of CTCF–DNA binding across the genome and as a result mediates CTCF-bound promoter–enhancer loops. More than colocalizing on active

promoter and enhancer regions in the mouse genome, analysis 10,544 ZNF143 motifs suggested a specific spacing of 37 bp between a ZNF143 and CTCF binding sites. Such a specific distance between two different TF binding motifs across the genome has been rarely reported, which led to the hypothesis that ZNF143 served as a regulator of CTCF by mediating its ability to bind to DNA. Loss of ZNF143 led to the loss of CTCF–DNA binding on these loci, mainly affecting CTCF-bound promoter–enhancer loops, but not TAD formation or

compartmentalization, consistent with global changes in gene expressions. Considering the ubiquitous expression and high homology of both ZNF143 and CTCF among mammals, we conclude that ZNF143 is a novel CTCF regulator that mediates CTCF–DNA binding on promoter and enhancer regions to form promoter–enhancer loops. However, we also noticed that a portion of affected loops are not associated with promoter regions (Supplementary Data 5), suggesting that ZNF143 might mediate other types of loops bound by CTCF, as CTCF is well known to play a critical role on insulators[34,35].

It is quite interesting that ZNF143 mainly regulates CTCF-bound promoter–enhancer loops but not TADs. That loss of ZNF143 only affects CTCF-bound promoter–enhancer loops suggests a novel subgroup of CTCF sites, regulated by ZNF143 and restricted to promoter and enhancer regions. It is widely reported that CTCF and cohesin are the major factors to maintain TAD formation across different cell types[4–9,15,16,21,26,36,37]. However, in contrast to conservation of TADs in different cell types, loop formations are far more dynamic and regulated not only by CTCF and cohesin, but other factors such as YY1[11] and chromatin remodeler Brg1[38], among others. Recent studies using high-resolution Micro-C revealed a large portion of promoter–enhancer loops under 200 bp resolution[39,40]. Interestingly, these new discovered promoter–enhancer interactions were often detected in the absence of CTCF and cohesin occupancy. Although their biological function remains to be explored, this finding suggests that further studies are required to better understand genome organization and its regulation. Our results in this study strongly suggest that ZNF143 mainly regulates fine-scale genome organization without altering higher-order chromatin structure.

Another interesting aspect of future study is regarding ZNF143 and CTCF motif distance and orientation. Since it has not been reported previously that the ZNF143 motifs lie precisely 37 bp from the nearest CTCF motif, it will be of interest in subsequent studies to investigate how this 37 bp distance mediates their co-localization. The DNA binding affinity of one TF can be affected by another factor binding nearby[25], so one hypothesis is that ZNF143 stabilizes CTCF–DNA binding when located 37 bp away. Another possibility is that ZNF143 acts as a pioneer factor to remodel chromatin status to make CTCF sites located 37 bp away accessible. Future experiments will reveal which hypothesis, or both, can account for the dependency of CTCF–DNA binding, and promoter–enhancer interactions, on ZNF143. On the other hand, in line with that CTCF motif orientation is critical for loop formation, how ZNF143 motif orientation contributes to looping will be of interest, as almost all ZNF143 and CTCF motifs located 37 bp apart from each other form the convergent orientation. It will be of interest to investigate whether changes in ZNF143 motif orientation will affect nearby CTCF binding, and hence affect looping. Of note, we do not have a clear answer on whether ZNF143 motifs are required on both anchors, and therefore additional studies with more detailed analysis will be necessary to address this question.

How chromatin organization mediates hematopoietic stem and progenitor cell integrity is an important aspect yet not fully explored. Previous studies found that complete loss of SMC3, one of the core cohesin ring components, caused bone marrow aplasia due to abnormal chromatid segregation, whereas *Smc3* haploinsufficiency increased self-renewal in culture and in vivo[41]. On the other hand, deletion of cohesin binding component STAG2 but not STAG1 increased HSC self-renewal and reduced differentiation capacity, suggesting that the cohesin complex plays different roles in HSPCs driven by different STAG subunits[42]. Other than the cohesin complex, coactivators and mediators were also reported to be critical for HSPC function through facilitating

active histone marks, such as H2A.Z/H3K27 acetylation on promoter/enhancer regions[43–45]. Although our studies of ZNF143 reveal another aspect of the regulation of HSPC chromatin organization through mediating the integrity of CTCF-bound loops, it is not clear that whether ZNF143 is critical for all HSPCs or only in LT-HSCs. As homo- or hetero-dimers could be formed within zinc- coordinating TFs, we anticipate that additional TFs will be discovered to contribute to chromatin structure and hence regulate HSPC integrity.

The model that ZNF143 mediates CTCF-bound promoter–enhancer loops may have relevance to interpreting previous studies. For example, it is reported that ZNF143 co-binds to approximately 40% of Notch1 sites in TLL[19,20]. Considering that Notch1 gain-of-function mutations are mostly observed in TLL patients, this might represent another example of CTCF dependency on another TF. Since disruption of this ZNF143-CTCF regulatory axis results in a severe anemia phenotype in both embryos and adults, in the agreement of ubiquitous expression and high homology of both ZNF143 and CTCF among mammals, the regulation of CTCF-bound loops controlled by ZNF143 might contribute to multiple physiologic and disease processes.

## Methods

**Mice**. C57BL/6J, *Mx1-Cre*[46], and *Vav1-iCre* mice[29] were obtained from The Jackson Laboratory; *Rosa26ERT2-Cre* mice[47] were obtained from Yoshiaki Ito. *Znf143* flox mice were generated by inserting loxP site before and after exon 1 of *Znf143*. *Znf143 f/f* mice were crossed with *Rosa26ERT2-Cre+* mice to generate *Znf143 f/f* x *Rosa26ERT2-Cre+* and *Znf143 f/f* x *Rosa26ERT2-Cre-* mice. Then *Znf143* deletion was achieved with tamoxifen injection as described in Figure S2. *Znf143 f/f* x *Vav1-iCre-* male mice were crossed with *Znf143 f/+* x *Vav1-iCre+* female mice to avoid germline deletion of the floxed allele. To study the function of *Znf143* in adult hematopoiesis, *Znf143 f/f* mice were crossed with *Mx1-Cre+* mice to generate *Znf143 f/f* x *Mx1-Cre+* and *Znf143 f/f* x *Mx1-Cre-* mice. Littermates of *Znf143 f/f* x *Mx1-Cre+* and *Znf143 f/f*; *Mx1-Cre-* mice were injected intraperitoneally (i.p.) with 300 μg poly I:C for three consecutive days. Then mice were euthanized at day 8 after the first injection to assess deletion. All mice were housed in a sterile barrier facility within the Comparative Medicine facility at the National University of Singapore under housing condition of 22 °C temperature, 50% humidity and a 12:12 light/dark cycle. All mice experiments performed in this study were approved by Institutional Animal Care and Use Committee of National University of Singapore. In this study, 8- to 12-week-old male mice were used bone marrow harvesting or transplantation; 12- to 16-week-old female mice were used for time mating. See "Methods" for more information.

**Antibodies and primers**. All antibodies and primers used in this study were listed in supplementary data 10 and 11. All full scans of Western-blot are provided in Supplementary Fig. 8.

**Fetal liver cell isolation**. For fetal liver isolation, *Znf143 f/+* x *Vav1-iCre+* male mice were mated with *Znf143 f/f* female mice in the evening on day 0. Embryos were collected at E14.5-day stage for fetal liver isolation and genotyping. Fetal livers were kept in PBS at 4 °C until genotyping of each embryo was completed. Fetal livers from *Znf143 f/+* x *Vav1-iCre-* (*Znf143 f/+*) and *Znf143 f/f* x *Vav1-iCre+* (*Znf143 −/−*) embryos were dissociated into single cells by passing through a 40 μm BD strainer with a 1 ml syringe plunger in PBS plus 2% FBS and 2 mM EDTA.

**Bone marrow cells (BMCs) isolation**. After $CO_2$ euthanasia, mice were dissected for vertebrae, femur, tibia, and hip collection. Bones were crushed using a pestle with ice-cold PBS and then bone marrow cells suspension were filtered through a 70 μm BD cell strainer. Red blood cells were removed with RBC lysis buffer treatment. After filtering, a 20 μl cell suspension was diluted into a 2 ml solution for white blood cell (WBC) counts using a NIHOKODEN auto blood cell counter under Pre-dilute 20 μl mode. The remaining cells were kept on ice for other experiments.

**Flow cytometry and fluorescence-activated cell sorting (FACS)**. For flow cytometry analysis, 1 million WBCs or Fetal liver cells were stained for 15 min as indicated in Supplementary Data 9. After staining, cells were washed with PBS once and then resuspended in 500 μl PBS for flow cytometry on an LSRII cytometer. Data collection was done with BD FACSDiva software (v8.0.1). The gating strategy for different populations is provided in Supplementary Fig 9.

For FACS sorting, mature blood cells were stained with biotin-labeled anti-lineage surface marker antibodies as indicated in Supplementary Data 9 for 30 min

incubation at 4 °C, followed by lineage depletion on an autoMACS machine under "depletes" mode.

Lineage depleted cells were stained as previously described for 15 min. Then cells were washed once and were resuspended to a final cell concentration of 40 M/ ml for FACS on an Arial I/II. Post-sort was performed for double checking of cell populations and for cell number counting. The gating strategy for different populations is provided in Supplementary Fig. 9.

FlowJo was used for data plot of flow cytometry and FACS.

**Transplantation**. Eight hundred thousand fetal liver cells (isolated from E14.5-day old *Znf143 f/+* x *Vav1-iCre-* or *Znf143 f/f* x *Vav1-iCre+* embryos) or RBC lysis buffer treated BMCs (from *Znf143 f/f* x *Mx1-Cre-* or *Znf143 f/f* x *Mx1-Cre+* male mice) were mixed with two hundred thousand RBC lysis buffer treated BMCs isolated from CD45.1 congenic mice to a final cell concentration of ten million per ml. Recipient CD45.1 congenic mice were lethally irradiated (9.5 Gy) using a γ-irradiator, followed by injecting one million mixed cells per mouse. Every four weeks after injection, 100 μl peripheral blood (PB) was collected from each mouse to check for chimerism. To examine chimerism formation, PB was lysed with 1 ml RBC lysis buffer for 5 min at room temperature. After washing with PBS, cells were stained with antibody mixtures as indicated in Supplementary Data 9 for 15 min at 4 °C, followed by analysis on an LSRII.

**RNA isolation, library construction, and next-generation sequencing**. The HSC SLAM population (LSK+CD150+CD48−) was sorted into PBS plus FBS and EDTA. After post-sort to confirm the sorted population and to count cell number, cells were spun down for RNA extraction following trizol-chloroform-ethanol extraction with glycogen or glycoblue precipitation. RNA samples with RNA integrity number (RIN) score higher than 8.0 were selected for RNA-seq library construction using SMART Seq v4 Ultra Low Input RNA kit from Takara and Nextera XT DNA library prep kit from Illumina, with PAGE-gel purification for final library size selection. For next-generation sequencing (NGS), 6 libraries (three from *Znf143 f/f* x *Rosa26ERT2-Cre-* mice, and three from *Znf143 f/f* x *Rosa26ERT2-Cre+* mice, all mice were littermates) were pooled together and were sequenced using a Nextseq high output 150 cycle v2 kit in the Illumina Nextseq 500 platform to obtain at least 50 million 76 bp paired-end reads per RNA library. Illumina bcl2fastq2 Conversion Software (v2.20) was used for Next Generation Sequencing library demultiplexing. Fastqc software was used for data quality check.

**Co-immunoprecipitation (co-IP)**. After post-sort to confirm the sorted population and to count cell numbers, HSPCs (lineage- c-kit + ) were washed with PBS once and then lysed in 1 ml IP buffer (50 mM Tris-Cl pH 8.0, 100 mM Na Fluoride, 30 mM Na Pyrophosphate, 2 mM Na Molybdate, 5 mM EDTA, 2 mM Na Vanadate, 1% NP-40, and freshly prepared protease inhibitor purchased from Merk, cat.no. 11697498001) on ice for 10 min with occasionally inverting. After spinning down at 13,200 rpm for 10 min at 4 °C, the supernatant was transferred into a new tube containing antibody pre-bound dynabeads protein beads for overnight incubation at 4 °C (anti-CTCF antibody, #07-729, Millipore; Anti-ZNF143 rabbit serum is generated in our lab); 50 μl of the supernatant was stored at -20 °C as input. The next day, after washing beads for five times with IP buffer, loading dye was added to the beads and input for boiling at 97 °C for 10 min, followed by Western-blot analysis.

**Chromatin immunoprecipitation (ChIP), library construction, and next-generation sequencing**. ChIP was performed as described previously[23]. Briefly, after post-sort to confirm the sorted population and to count the cell numbers, HSPCs (lineage- c-kit+) were resuspended in PBS to a concentration of 1 M per ml. Cells were fixed with 1% formaldehyde for 15 min at RT with rotation. Then formaldehyde was neutralized by adding 2.5 M glycine to a final concentration of 0.125 M and rotating for 5 min at RT. After washing with PBS, cells were lysed with ChIP SDS lysis buffer (100 mM NaCl, 50 mM Tris-Cl pH8.0, 5 mM EDTA, 0.5% SDS, 0.02% NaN₃, and fresh protease inhibitor), and then stored at -80 °C until further processing. Nuclei were collected by spinning down at 13,000 rpm for 10 min. The nuclear pellet was resuspended in IP solution (2 volume ChIP SDS lysis buffer plus 1 volume ChIP triton dilution buffer (100 mM Tris-Cl pH8.6, 100 mM NaCl, 5 mM EDTA, 5% Triton X-100), and fresh proteinase inhibitor) for soni-cation using either a Bioruptor or Covaris to obtain 200 bp to 500 bp DNA fragments. After spinning down to remove debris, sonicated chromatin was pre-cleared by adding 30 μl washed dynabeads protein A/G and rotated at 4 °C for 2 h. Pre-cleared chromatin was incubated with antibody pre-bound dynabeads protein A/G overnight at 4 °C (Anti-ZNF143 rabbit serum is generated in our lab; anti-CTCF antibody, # 07-729, Millipore; anti-tri-methyl-histone H3 (Lys27) antibody, #9733BC, Cell Signaling Technology; anti-histone H3 (acetyl K27), #ab4729, Abcam). Next day, magnetic beads were washed through the following steps: buffer 1 (150 mM NaCl, 50 mM Tris-Cl, 1 mM EDTA, 5% sucrose, 0.02% NaN₃, 1% Triton X-100, 0.2% SDS, pH 8.0) two times; buffer 2 (0.1% deoxycholic acid, 1 mM EDTA, 50 mM HEPES, 500 mM NaCl, 1% Triton X-100, 0.02% NaCl, pH 8.0) two times; buffer 3 (0.5% deoxycholic acid, 1 mM EDTA, 250 mM LiCl, 0.5% NP40, 0.02% NaN₃) two times; TE buffer one time. After reverse crosslinking and pur-ification of DNA, qPCR or library construction was performed accordingly.

For library construction, a Thruplex DNA seq kit from Rubicon Genomics was used following the manufacturer's instructions. TBE PAGE-gel size selection was performed for the final library size selection to obtain 250 bp to 500 bp ChIP-seq libraries. Pooled libraries at 10 nM concentration were sequenced in the Illumina Nextseq 500 platform. Illumina bcl2fastq2 Conversion Software (v2.20) was used for Next Generation Sequencing library demultiplexing. The Fastqc software was used for data quality check.

**ChIP-reChIP**. ZNF143 ChIP was performed as described above until the TE buffer washing step. Then immunocomplexes were eluted by incubating with 50 μl 10 mM DTT for 30 min at RT. After diluting the elution with 1 ml IP buffer, 20 μl CTCF antibody or 5 μg normal rabbit IgG (#2729, Cell Signaling Technology) pre-bound protein A dynabeads were added for overnight rotation at 4 °C, followed by standard ChIP procedure.

**In situ HiC library construction**. HSPCs (lineage- Sca-1+ c-kit+) were used for in situ HiC libraries construction using Arima-HiC kit and Swift biosciences® Accel-NGS® 2S plus DNA library kit following the low input procedure instructions.

**Chromatin conformation capture (3C)**. 3C was performed as described pre-viously[48]. In brief, HSPCs (lineage- c-kit+) were crosslinked in 1% formaldehyde for 15 min at RT with rotating. Then formaldehyde was neutralized by adding 2.5 M glycine to a final concentration of 0.125 M and rotating for 5 min at RT. After washing in PBS, cells were resuspended in 1 ml lysis buffer (10 mM Tris-HCl pH8.0, 10 mM NaCl, 5 mM EDTA, 0.5% NP 40, add protease inhibitor freshly) and lysed with a douncer on ice. After spinning down at 2000 rpm for 10 min at 4 °C, nuclei were washed with 1X CutSmart buffer from NEB, followed by resuspending in 1.8 ml nuclease free (NF) H₂O. After splitting into 4 tubes containing 60 μl of 10X CutSmart buffer and 15 μl of 10% SDS buffer, samples were incubated at 37 °C 1 h with shaking, followed by adding 75 μl of 20% Triton X-100 to each tube for 1-h incubation at 37 °C with shaking. From each tube 20 μl samples were taken out and were combined as "undigested" samples and stored at -20 °C. HindIII-HF was added into each tube (700 U per tube) for overnight digestion at 37 °C with shaking. The next day after confirming a digestion efficiency over 80%, 50 μg digested DNA were taken out into a new tube and the volume adjusted to 600 μl with NF H₂O. After adding 80 μl 10% SDS, samples were incubated at 65 °C for 25 min. Heat inactivated chromatin was added into ligation mix (5 ml 10X ligation buffer, 2.687 ml 20% Triton X-100) and the volume adjusted to 50 ml with NF H₂O to a final concentration of 1 ng/μl, followed by 1 h incubation at 37 °C with rotation at 30 RPM. After adding 990 U T4 DNA ligase (Thermo Scientific), samples were incubated at 16 °C overnight with rotation at 15 RPM. The next day, 132 μl of proteinase K (Ambion) were added into each sample, followed by 65 °C incubation overnight. The next day, after adding 132 μl of RNase A (PureLink), samples were incubated at 37 °C for an hour, followed by phenol: chloroform DNA purification. Chromatin-chromatin interactions between the *Cebpa* promoter and −37 kb enhancer region and between the *Cebpa* promoter and −18 kb upstream region were examined by qPCR. The −76 kb upstream region having no interaction with *Cebpa* promoter was selected as negative control.

**Data analysis**

*Dot plots, survival curves*. All dot plots and survival curves are drawn with *GraphPad Prism* unless further indicated in the data analysis section.

**RNA-seq data analysis**. After trimming of adaptor sequences with *TrimGalore*, paired-end reads were aligned to mouse genome GRCm38 (mm10) using *STAR*[49]. After calculating raw gene expression (raw reads count) by *FeatureCounts*[50], normalized profiling results were obtained with *DESeq2* for differential expression analysis[51]. Gene Set Enrichment Analysis (GSEA) was performed using the *GSEA* pipeline from the Broad Institute[52,53]. For visualization on the UCSC genome browser or IGV, bam files obtained from *STAR* were first converted into bedgraph format using *BEDTools*[54], then converted into bigwig files with UCSC bed-GraphtoBigWig program.

**ChIP-seq data analysis**. After trimming of adaptor sequences by *TrimGalore*, paired-end reads were aligned to mouse genome GRCm38 (mm10) using *bowtie2* with output in sam file format[55]. *SAMtools* was used (1) to convert sam file to bam file, (2) to remove both low mapping quality score reads (mapq score less than 20) and PCR duplicates[56]. ChIP-seq peaks were called by *MACS2* with a false discovery rate (FDR) less than 0.001, followed by peak annotation with *HOMER*[57,58]. The resulting alignments were extended to 150 bp and then converted into signals by *BEDTools* in bedGraph format[54]. We adjusted the average coverage of each ChIP-seq experiment to 1. Finally, the resulting files were converted to bigwig format by *bedGraphToBigWig* for visualization and generating heatmaps[59]. Correlation of replicates was assessed with the *Deeptools* multiBigwigSummary BED-file function[60].

**Motif discovery**. To discover CTCF and ZNF143 motifs in the mouse genome, bed files generated by *MACS2* were used for de novo motif scanning using *HOMER* findMotifsGenome function with default settings[58]. Then the discovered CTCF and ZNF143 motifs were used for whole genome scanning using *HOMER* scanMotif-GenomeWide function with output of bed files. Distances between CTCF and ZNF143 motifs in genome-wide were determined by *BEDTools*. R package ggplot was applied for result visualization in *Rstudio*.

ZNF143 mutant motifs were generated by loading ZNF143 motif obtained from *HOMER* to the online program *Regulatory Sequence Analysis Tools (RSAT)* with permute-matrix function[24]. Then distances between CTCF and mutant ZNF143 motifs were discovered by *BEDTools* and were visualized by R in *Rstudio*.

*Motifs orientation*. As ZNF143 and CTCF motifs are not palindromic and could locate on either the forward or reverse strand of DNA, a pair consisting of ZNF143 and the nearest CTCF motif have four potential orientations as illustrated in Figure S2C, lower panel: (a) convergent on opposite strands, (b) divergent on opposite strands, (c) both on forward strand (F-F), (d) both on reverse strand (R-R). To determine orientations in 10,544 pairs of ZNF143 and CTCF motifs located 37 bp apart, strand specificity of each motif was detected using *HOMER*, and then each pair of motifs were sub-grouped as described above to get the final count by R in *Rstudio*.

**Promoter and enhancer calling**. Active promoters were identified by H3K27ac peaks within 1000 bp to a nearest transcription start site (TSS). Active enhancers were identified by H3K27ac peaks that located outside of promoter regions.

**Quantification of ChIP-seq signal on promoter, enhancer, or loops anchor regions**. For calculating the average of ChIP-seq signals on a promoter, enhancer, or nearby CTCF motifs within loops anchor regions, *HOMER* was used to make tag directories of individual ChIP-seq bam files, followed by calculating ChIP-seq signal coverage using *HOMER* annotatePeaks function[58]. The output was visualized as a mean plot using the *R* package in *Rstudio*.

**Heatmap of ChIP-seq signal on promoter, enhancer, or loops anchor regions**. *Deeptools* or *ngsplot* was used to generate heatmap of ChIP-seq on different regions[60,61]. First, desired bed files indicating such regions were generated as described in this manuscript. Then bigwig files from ChIP-seq data were used to generate a matrix on input regions by *Deeptools* computeMatrix function, followed by *Deeptools* plotHeatmap function to draw heatmap. *Deeptools* was also used for calculating genome-wide correlations among ChIP-seq signals in bigWig files with bin size of 5000 bp.

For *ngsplot*, bam files from ChIP-seq data and desired bed files were used for heatmap generation with ngs.plot.r function.

**HiC data analysis**. HiC contact matrices were generated using the *Juicer tools suite* (*v1.9.9_jcuda.0.8*) following the instruction[7]. The Knight-Ruiz balanced observed/expected interaction frequencies were calculated at 2.5 Mb, 1 Mb, 500 kb, 250 kb, 100 kb, 50 kb, 25 kb, and 5 kb resolution. The matrices were visualized in *Juicebox* (*v1.9.8*). The eigenvector values were calculated at 1 Mb resolution using *Juicer tools* eigenvector function. The PC1 values were calculated using *HOMER* following its HiC analysis tutorial[62].

Insulation scores were calculated as described[28]. Dense contact matrices for each chromosome were extracted from .hic files using *Juicer tools* dump function returning the observed contacts without normalization at 25 kb resolution. The resulting matrices were transformed into the *HiC-PRO* format[63] using a custom *R* script. The insulation score was calculated from these *HiC-PRO* contact matrices using the the matrix2insulation.pl function in *HiC-PRO*[28]. The resulting insulation score per chromosome was aggregated into a single bedGraph and then transformed to bigwig using the *bedGraphToBigWig* function from UCSC together with the chromosome size information for mm10 in UCSC. Profiles of genome-wide insulation score around TADs boundaries were visualized by *Deeptools*.

TADs were called using the *Juicer tools* Arrowhead algorithm at 25 kb resolution. Loops were called using *Juicer tools* Hiccups function at 5 kb resolution. Loops intensities were calculated by *Juicer tools* dump function with input of .hic files generated by *Juicer tools*. Aggregate peak analysis on subgroup loops was performed using *Juicer tools* APA function, and the result is visualized with *R* package in *Rstudio*. To discover CTCF motifs on loop anchors, bedpe files containing loops information were examined by *Juicer tools* MotifFinder function.

**Reporting summary**. Further information on research design is available in the Nature Research Reporting Summary linked to this article.

## Data availability
The ChIP-seq, RNA-seq, and HiC data that support the findings of this study have been deposited in GEO with the GEO accession number "GSE144712". Biological material used in this study can be obtained from the authors upon request. The authors declare that all other data supporting the findings of this study are available within the paper and its supplementary information files. Source data are provided with this paper.

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

## Acknowledgements

The authors thank Michelle Mok Meng Huang and Chelsia Qiuxia Wang for assistance with flow cytometry experiments. This research was supported by the Singapore Ministry of Health's National Medical Research Council under its Singapore Translational Research (STaR) Investigator Award, and by the National Research Foundation Singapore and the Singapore Ministry of Education under its Research Centres of Excellence initiative, as well as NIH grants 1R35CA197697 and P01 HL131477 to D.G.T., and Xiu Research Fund to L.C.

## Author contributions

D.G.T initiated the project and provided guidance throughout. D.G.T, Q.Z, S.L, and M.Y designed the experiments. Q.Z planned and performed molecular, cellular, and mouse experiments. M.G. and Z.H.T participated in mouse experiments. Q.Z, R.T.M, B.L, and L.K. performed bioinformatics analysis. Q.Z, M.Y, M.J.F, M.O, T.B, B.R, and D.G.T interpreted the data. Q.Z. and D.G.T. wrote the manuscript with input from M.Y, L.C, A.N, and B.R.

## Competing interests

B.R. is a co-founder of Arima Genomics, Inc. All other authors declare no competing interests.
