## [Peer Review File · Nature Communications]

REVIEWER COMMENTS

Reviewer #1 (Remarks to the Author):

Zhou et al report the function of ZNF143 in partnership with CTCF in the formation of chromatin interactions between cis-regulatory elements in mouse through a series of assays dependent on ZNF143 expression modulation. Focusing on the hematopoietic system, they show that ZNF143 is required for the maintenance of hematopoietic stem and progenitor cell integrity.

Overall, this is an important study that significantly contributes to the existing literature on mechanisms regulating chromatin interactions. The discoveries presented in this manuscript on the role of ZNF143 on hematopoietic stem and progenitor cells is highly interesting to the stem and developmental biology communities.

Minor comments:

1. Intro page 3, Consider adding Rao et al (DOI:<https://doi.org/10.1016/j.cell.2014.11.021>) as a reference supporting the following statement "Despite bioinformatic analysis suggesting that, similar to cohesion, ZNF143 col-localizes with CTCF". Consider adding this same reference to the sentence on line 85 of page 4.
2. Intro page 3, same sentence as point 1: Reference 22 demonstrated that ZNF143 directly regulates chromatin interactions. Reporting this information would strengthen the elements of this manuscript revealing a role for ZNF143 in regulating chromatin interactions.
3. Regarding ZNF143 depletion in HSPC, consider adding a description of the cell-type specificity of the Vav1-iCre and the Mx1-Cre models when introducing them in the results section.
4. Current results support a role for ZNF143 in the maintenance of hematopoietic stem and progenitor integrity. This suggests that ZNF143 is either critical across all HSPCs or solely at the apex of the hierarchy. While Cre models might not allow to discriminate these two hypotheses, the authors may wish to discuss them in their manuscript.
5. In line with comment 4, I would welcome a section of the discussion to be committed to deepening the conversation on the role of chromatin interaction regulators in the maintenance of stem and progenitor cells integrity.
6. In the discussion, the conclusion that CTCF binding to the chromatin is dependent on ZNF143 should be discussed in contrast to the model presented in reference 22, where each factor is suggested to bind to opposite ends of chromatin interactions.
7. The authors may want to consider updating their title to highlight the importance of their work toward the maintenance of hematopoietic stem and progenitor cells integrity.

Reviewer #3 (Remarks to the Author):

Previous studies have shown that ZNF143 is enriched at chromatin loop anchors and that ZNF143 is required for formation of chromatin loops at individual loci. This manuscript used conditional ZNF143-knockout mice and applied HiC to show a global role of ZNF143 in regulating chromatin loops. In addition, the authors show novel mechanisms underlying the function of ZNF143, such as its role in stabilizing CTCF binding at promoters and enhancers. The manuscript was well written and easy to follow. Figures were made in high quality. Some of my suggestions are:

1. Figure 1A: annotation of CTCF-ZNF143 shared, CTCF-alone and ZNF143-alone regions is missing. Are they promoters, enhancers, or insulators/TAD boundaries? Same comments for Figure 5C. It would be key to show that the ZNF143-affected CTCF bindings are enriched in promoter/enhancers instead of TAD anchors/insulators.
2. It is very interesting that CTCF and ZNF143 motifs are 37bp apart from each other and that ZNF143 affects CTCF binding. Does CTCF also influence ZNF143 binding? A few ChIP-qPCR on some of the shared sites should be able to answer the question.

3. Orientation of CTCF motifs at its binding sites has been shown as a key factor for chromatin loop formation. Is there any pattern for ZNF143 motif orientation at the loop anchors?
4. Annotation of loops affected by ZNF143 is missing in the Figures. Are these loops connecting Promoter-Enhancer, E-E, P-P, Insulator-P, I-E, or I-I? Such information is key in my mind, as the authors claim that ZNF143 is important for promoter-enhancer loops.
5. Figure 2E shows an example of ZNF143 deletion leads to a loss of interaction between two sites that loses CTCF. Are these loci promoters, enhancers etc? Also, I recommend the authors present more individual examples like 2E in supplementary figures.
6. The authors suggest that ZNF143 regulates CTCF bindings, which in turn regulates chromatin loops. Is there a global assessment regarding how ZNF143-mediated CTCF bindings associate with the ZNF143-mediated chromatin loops?
7. Figure 2F: the 3C assay seems missing negative controls (regions that doesn't interact with the promoter region of Cebpa)
8. The Cebpa locus should have figures presenting ChIP-seq and HiC tracks (like 2E) that show the chromatin environment.
9. It is unclear whether or not a chromatin loop requires CTCF/ZNF143 at both anchors or just one. The model (Figure 5E) presented by the authors show that both anchors need CTCF/ZNF143, but this analysis seems missing in the Figures.

Minor:

1. It should be noted in the introduction that the previous study from Bailey, 2015 have shown that ZNF143 is essential for formation of chromatin loops at individual loci (3C assays).
2. The order of the figures should match the order of the text. Figure 3 is mentioned earlier than Figure 2F-G in the text.

Reviewer #4 (Remarks to the Author):

Zhou et al generate a ChIP-grade polyclonal antibody against ZNF143 and ZNF143 f/f mice and perform an elegant series of experiments in ZNF f/f of ZNF f/+ and ZNF -/- mouse HSPCs including ZNF143/CTCF/H3K27Ac ChIP-seq, ZNF143/CTCF re-ChIP, ZNF143/CTCF co-IP, Hi-C, mRNA transcriptomics, E14.5FL flow cytometry and bone marrow transplantation and PB donor chimerism to show that ZNF143/CTCF interactions are common at active promoters/enhancers and loss of ZNF143 results in loss of chromatin loops and gene expression changes within TADs that remain largely unchanged. These changes in mouse HSPCs are associated with changes in the expression levels of key drivers of HSC maintenance/differentiation consistent with the reported phenotypic changes. Overall, the data is clearly presented and convincing.

Comments

1. The focus of the study is mouse HSPCs. The title should reflect this.
2. The numbers of regions bound by ZNF143/CTCF in Fig 1A should be given.
3. Do regions where CTCF is bound with no ZNF143 binding (Figure 1A) correspond to TADs?
4. Any lymphocytes amongst the CD45.2 donor cells in Fig 4?
5. CTCF clearly binds DNA without ZNF143 and as the authors data show maintain TADs in its absence in mouse HSPCs- some discussion about this and the absence of change with regards to this higher order chromatin organisation would be welcome.

Point-by-point response:

Reviewer #1 (Remarks to the Author):

Zhou et al report the function of ZNF143 in partnership with CTCF in the formation of chromatin interactions between cis-regulatory elements in mouse through a series of assays dependent on ZNF143 expression modulation. Focusing on the hematopoietic system, they show that ZNF143 is required for the maintenance of hematopoietic stem and progenitor cell integrity.

Overall, this is an important study that significantly contributes to the existing literature on mechanisms regulating chromatin interactions. The discoveries presented in this manuscript on the role of ZNF143 on hematopoietic stem and progenitor cells is highly interesting to the stem and developmental biology communities.

~~~~~

**Reviewer #1, Minor Comment #1:**

Intro page 3, Consider adding Rao et al (DOI:<https://doi.org/10.1016/j.cell.2014.11.021>) as a reference supporting the following statement ?Despite bioinformatic analysis suggesting that, similar to cohesin, ZNF143 col-localizes with CTCF?. Consider adding this same reference to the sentence on line 85 of page 4.

**Response to Reviewer #1, Minor comment #1:**

We thank reviewer for this comment. We have added this reference to these two places accordingly in the introduction and the results (highlighted now Reference 7).

**Line 63, Introduction:**

Despite bioinformatic analysis suggesting that, similar to cohesin, ZNF143 co-localizes with CTCF7,21,22, ...

**Line 87, Results:**

..., even though it is reported that ZNF143 and CTCF are both chromatin structure factors7,21,22

~~~~~  
Reviewer #1, Minor Comment #2:

Intro page 3, same sentence as point 1: Reference 22 demonstrated that ZNF143 directly regulates chromatin interactions. Reporting this information would strengthen the elements of this manuscript revealing a role for ZNF143 in regulating chromatin interactions.

Response to Reviewer #1, Minor Comment #2

We appreciate reviewer's suggestion. We have included following sentence in the introduction report this information:

Line 63, Results:

Despite bioinformatic analysis suggesting that, similar to cohesin, ZNF143 co-localizes with CTCF^{7,21,22}, and knocking down ZNF143 affects chromatin interaction at individual loci²², how CTCF and ZNF143 regulate the function of each other remains unknown.

~~~~~  
**Reviewer #1, Minor Comment #3:**

Regarding ZNF143 depletion in HSPC, consider adding a description of the cell-type specificity of the *Vav1-iCre* and the *Mx1-Cre* models when introducing them in the results section.

**Response to Reviewer #1, Minor Comment #3:**

We thank reviewer for this comment. We have added following sentences in the results section to describe the cell-type specificity of these two models:

Line 192, Results:

The *Vav1-iCre* mouse model is widely used for studies of hematopoiesis in embryonic development as it starts expressing at E9.5 and is greatly enhanced at E12.5 in hematopoietic cells27.

Line 202, Results:

..., and reduction of ZNF143 protein was detected in HSPCs 8 days after the first poly I:C injection (Fig. S1G).

~~~~~

Reviewer #1, Minor Comment #4:

Current results support a role for ZNF143 in the maintenance of hematopoietic stem and progenitor integrity. This suggests that ZNF143 is either critical across all HSPCs or solely at the apex of the hierarchy. While Cre models might not allow to discriminate these two hypotheses, the authors may wish to discuss them in their manuscript.

Response to Reviewer #1, Minor Comment #4:

We agree that our current results are unable to discriminate whether ZNF143 is critical across all HSPCs or solely in LT-HSCs. We have included following paragraph in the discussion to acknowledge this.

Line 303, Discussion:

How chromatin organization mediates hematopoietic stem and progenitor cell integrity is an important aspect yet not fully explored. Previous studies found that complete loss of SMC3, one of the core cohesin ring components, caused bone marrow aplasia due to abnormal chromatid segregation, whereas *Smc3* haploinsufficiency increased self-renewal in culture and *in vivo*³⁹. On the other hand, deletion of cohesin binding component STAG2 but not STAG1 increased HSC self-renewal and reduced differentiation capacity, suggesting that the cohesin complex plays different roles in HSPCs driven by different STAG subunits⁴⁰. Other than the cohesin complex, coactivators and mediators were also reported to be critical for HSPC function through facilitating active histone marks, such as H2A.Z/H3K27 acetylation on promoter/enhancer regions⁴¹⁻⁴³. Although our studies of ZNF143 reveal another aspect of the regulation of HSPC chromatin organization through mediating the integrity of CTCF-bound loops, it is not clear that whether ZNF143 is critical for all HSPCs or only in LT-HSCs. As homo- or hetero-dimers could be formed within zinc-coordinating transcription factors, we anticipate that additional transcription factors will be discovered to contribute to chromatin structure and hence regulate HSPC integrity.

~~~~~

**Reviewer #1, Minor Comment #5:**

In line with comment 4, I would welcome a section of the discussion to be committed to deepening the conversation on the role of chromatin interaction regulators in the maintenance of stem and progenitor cells integrity.

**Response to Reviewer #1, Minor Comment #5:**

We thank reviewer for this suggestion. We have included such discussion in the response to comment 4.

~~~~~

Reviewer #1, Minor Comment #6:

In the discussion, the conclusion that CTCF binding to the chromatin is dependent on ZNF143 should be discussed in contrast to the model presented in reference 22, where each factor is suggested to bind to opposite ends of chromatin interactions.

Response to Reviewer #1, Comment #6:

We thank reviewer for pointing out the different model proposed by us and in Reference 22. However, we consider our model does not contrast to theirs, but is rather a further extension for the following reasons:

(1) In both our results, enrichment of ZNF143 is detected at CTCF-bound sites (Figs. 1A, 1D-1F, S2E, S2F, S2G; and Reference 22, Fig. 1D, lower panel). Furthermore, we demonstrate that ZNF143 and CTCF interact with each other (Fig. 1B) and then co-occupy the same genome regions (Fig. 1F), indicating that ZNF143 and CTCF indeed sit together at the same regions across both human and mouse genomes.

(2) Moreover, when investigating CTCF binding changes after loss of ZNF143, we detect a significant decrease of CTCF binding on ZNF143 cobinding sites (Fig. 1A), without affecting CTCF mRNA and protein levels (Fig. S2A, S2B), demonstrating that CTCF binding to the chromatin is dependent on ZNF143.

(3) On the other hand, we observed that in Reference 22, analysis of GM12878 ChIP-seq results suggests that ZNF143 is highly enriched on promoter regions, where modest binding of CTCF is detected (Reference 22, Fig. 1D upper panel), whereas in our results, strong enrichment of ZNF143 and CTCF are

detected on promoter regions in HSPCs (Fig. 1D). However, we also notice that other studies demonstrate CTCF significantly enriches on promoter regions in other cell types (DOI: 10.1016/j.molcel.2019.08.015, Fig. 1A, 1C). We hypothesize that CTCF binding on promoter regions might vary in different cell types, which could account for the differences in our model and that presented in Reference 22.

We have included the following sentence in the discussion to address this:

Line 255, Discussion:

A previous study suggested that zinc finger transcription factor ZNF143 is involved in promoter-enhancer loops²²; however, how it regulates such loops is unknown. In this study, we further investigate the function between ZNF143 and CTCF and demonstrate that...

~~~~~

**Reviewer #1, Minor Comment #7:**

The authors may want to consider updating their title to highlight the importance of their work toward the maintenance of hematopoietic stem and progenitor cells integrity.

**Response to Reviewer #1, Minor Comment #7:**

We thank reviewer for the comment. We have updated the title to the following to address this:

ZNF143 mediates CTCF-bound promoter-enhancer loops required for murine hematopoietic stem and progenitor cell function

~~~~~

Reviewer #3 (Remarks to the Author):

Previous studies have shown that ZNF143 is enriched at chromatin loop anchors and that ZNF143 is required for formation of chromatin loops at individual loci. This manuscript used conditional ZNF143-knockout mice and applied HiC to show a global role of ZNF143 in regulating chromatin loops. In addition, the authors show novel mechanisms underlying the function of ZNF143, such as its role in stabilizing CTCF binding at promoters and enhancers. The manuscript was well written and easy to follow. Figures were made in high quality. Some of my suggestions are:

Reviewer #3, Comment #1:

Figure 1A: annotation of CTCF-ZNF143 shared, CTCF-alone and ZNF143-alone regions is missing. Are they promoters, enhancers, or insulators/TAD boundaries? Same comments for Figure 5C. It would be key to show that the ZNF143-affected CTCF bindings are enriched in promoter/enhancers instead of TAD anchors/insulators.

Response to Reviewer #3, Comment #1:

We thank reviewer for the comment. We have now included the annotation of these different subgroups of peaks for Figure 1A and Figure 5C in new Figure 1B, Figure S3D, and Figure S5F. These annotations indicate that approximately 90% of CTCF-ZNF143 shared peaks are located on promoter/enhancer regions (Figure S2D), whereas within 3,086 TADs detected in *Znf143* *f/f* cells, 2,331 CTCF-alone peaks are detected on TADs boundaries (Figure S4D). Similarly, around 80% of ZNF143 related CTCF peaks located on loop anchors are also located on promoter/enhancer regions, whereas less than 20% of ZNF143 unrelated peaks locate on these regions (Figure S7F). This annotation provides more confidence that ZNF143 specifically regulates CTCF-bound loops and not TADs.

Figure S2D

Figure S2D legend:

(D) Bar chart describing peak distribution of different subgroups of CTCF and ZNF143 peaks on promoter or enhancer regions. CTCF and ZNF143 ChIP-seq peaks detected in *Znf143* *ff* cells are sub-grouped into three groups (CTCF-ZNF143 shared, CTCF alone, or ZNF143 alone), and the percentage of peaks located on promoter or enhancer regions in each subgroup is presented.

Line 118, Results:

Since approximately 90% of ZNF143 and CTCF co-bound regions were on promoter/enhancer regions (Figures S2D, S2G), we wondered whether ZNF143 is involved in CTCF-mediated transcriptional regulation.

Figure S4D

Figure S4D legend:

(D) Bar chart presenting the number of peaks located on TADs boundaries. CTCF and ZNF143 peaks detected in *Znf143* *ff* cells were sub-grouped as described in Figure S2D, and the number of peaks located on TADs boundaries in the different groups is presented.

Line 183 Results:

In line with that observation, 2,331 CTCF-alone peaks and less than 400 CTCF-ZNF143 shared peaks were detected on TAD boundaries (Figure S4D), indicating that ZNF143 is a novel factor mediating CTCF-bound loops but not TADs.

Figure S7F

Figure legend

(F) Bar chart describing peak distribution of different subgroups of CTCF peaks on promoter or enhancer regions. CTCF peaks are sub-grouped as described in Figure 5C, and the percentage of peaks located on promoter or enhancer regions in each subgroup is presented.

~~~~~

**Reviewer #3, Comment #2:**

It is very interesting that CTCF and ZNF143 motifs are 37bp apart from each other and that ZNF143 affects CTCF binding. Does CTCF also influence ZNF143 binding? A few ChIP-qPCR on some of the shared sites should be able to answer the question.

**Response to Reviewer #3, Comment #2:**

We appreciate the insightful question. We have included global ZNF143 ChIP-seq analysis on ZNF143-CTCF shared sites and ZNF-alone sites in *Znf143* *ff* cells in new Figure S2E. The result indicates that compared to ZNF-alone sites, the average signal on ZNF143-CTCF shared sites is much stronger, suggesting CTCF also influences ZNF143 binding.

Figure S2E

Figure legend

(E) The mean plot of ZNF143 ChIP-seq within  $\pm 4$  kb from peak centers. ZNF143 peaks detected in *Znf143 f/f* cells were divided into 1) ZNF143-CTCF overlapped, and 2) ZNF143-alone. The X axis represents the relative distance to the peak center, while the Y axis represents the mean of normalized read counts of the ZNF143 ChIP-seq signal in each group.

Line 112, Results:

On the other hand, we also queried whether CTCF facilitates ZNF143 binding. We observed higher enrichment of ZNF143 on ZNF143-CTCF shared sites compared to ZNF143-alone sites, suggesting that ZNF143 and CTCF regulate binding of one another (Figure S2E).

**Reviewer #3, Comment #3:**

Orientation of CTCF motifs at its binding sites has been shown as a key factor for chromatin loop formation. Is there any pattern for ZNF143 motif orientation at the loop anchors?

**Response to Reviewer #3, Comment #3:**

We thank reviewer for this important question. As illustrated in new Figure S2C lower panel, after examining four potential types of orientation between ZNF143 and the nearest CTCF motifs, we found that within 10,544 ZNF143 motifs located 37 bp away from the nearest CTCF motifs, 10,529 motifs (99.9%) formed the convergent orientation with the nearest CTCF motifs. We have included this assessment in the

new Figure 1C. With such a large percentage of ZNF143 motifs adjacent to a CTCF motifs in a specific orientation, we hypothesize that ZNF143 mediates CTCF-bound loops with a specific binding pattern. We have included this new data in the result and discussion section.

Figure S2C, lower panel

Figure S2C legend:

Lower panel: Illustration of four possible patterns of motifs orientation between ZNF143 and the nearest CTCF motifs: 1) convergent on opposite strands (convergent); 2) divergent on opposite strands (divergent); 3) both on forward strand (F-F); 4) both on reverse strand (R-R).

Figure 1C

Figure 1C legend:

(C) Histogram describing the distance between CTCF and ZNF143 motifs on the murine genome (mm10). The X axis represents the relative distance from the ZNF143 motif to the CTCF motif, while the Y axis represents the number of ZNF143 motifs in each bin. Each bin size is 1 bp. The table presents number of ZNF143 motifs in the different orientation patterns to the nearest CTCF motifs within 10,544 ZNF143 motifs located 37 kb apart from the nearest CTCF motifs.

Line 102, Results:

Since it is known that the orientation of CTCF motifs is a key factor for chromatin loop formation7,8,26,27, we were wondering whether ZNF143 and CTCF motifs were oriented in a pattern. Four possible patterns exist between these two motifs as they are not palindromic: ZNF143 and CTCF motifs are 1) convergent on opposite strands (convergent); 2) divergent on opposite strands (divergent); 3) both on forward strand (F-F); 4) both on reverse strand (R-R). After analyzing 10,544 pairs of ZNF143 and CTCF motifs located 37 bp away from each other, we observed 99.9% (10,529) formed the convergent orientation (Figures 1C, S2C). With such a remarkably consistent pattern between ZNF143 and CTCF motifs across genome, we hypothesized that ZNF143 and CTCF work together genome-wide for loop formation and gene expression.

Line 296, Discussion:

On the other hand, in line with that CTCF motif orientation is critical for loop formation, how ZNF143 motif orientation contribute to looping will be of interest, as almost all ZNF143 and CTCF motifs located 37 bp apart from each other form the convergent orientation. It will be of interest to investigate whether changes in ZNF143 motif orientation will affect nearby CTCF binding, and hence affect looping.

Line 652, Methods:

Motifs orientation

As ZNF143 and CTCF motifs are not palindromic and could locate on either the forward or reverse strand of DNA, a pair consisting of ZNF143 and the nearest CTCF motif have four potential orientations as illustrated in Figure S2C, lower panel: a. convergent on opposite strands, b. divergent on opposite strands, c. both on forward strand (F-F), d. both on reverse strand (R-R). To determine orientations in 10,544 pairs of

ZNF143 and CTCF motifs located 37 bp apart, strand specificity of each motif was detected using *HOMER*, and then each pair of motifs were sub-grouped as described above to get the final count by *R*.

~~~~~

Reviewer #3, Comment #4:

Annotation of loops affected by ZNF143 is missing in the Figures. Are these loops connecting Promoter-Enhancer, E-E, P-P, Insulator-P, I-E, or I-I? Such information is key in my mind, as the authors claim that ZNF143 is important for promoter-enhancer loops.

Response to Reviewer #3, Comment #4:

We thank reviewer for this comment. We have added the requested annotation to the new Table S10 and have rephrased the results (see below) and added to the discussion. The results indicate that within these loops, 38% are promoter-enhancer loops, and 10.7% are promoter-promoter loops, suggesting that indeed ZNF143 is important for promoter-enhancer loops. However, we cannot exclude the possibility that ZNF143 is also involved in other types of loops associated with CTCF.

Line 142, Results:

Further annotation of these affected loops revealed that 38% of them mediated promoter-enhancer and 10.7% mediate promoter-promoter interaction, suggesting that the loss of ZNF143 mainly affected promoter-enhancer loops (Figures 2A, 2B, Table S10).

Table S10 annotation of ZNF143 affected loops

	peak number	percentage
total ZNF143 affected loops	4013	100%
promoter-enhancer loops	1524	38.00%
promoter-promoter loops	429	10.70%
enhancer-enhancer loops	200	4.98%
Other types of loops	1860	46.35%

Line 269, Discussion:

However, we also noticed that a portion of affected loops are not associated with promoter regions (Table S10), suggesting that ZNF143 might mediate other types of loops bound by CTCF, as CTCF is well known to play a critical role on insulators^{34,35}.

Reviewer #3, Comment #5:

Figure 2E shows an example of ZNF143 deletion leads to a loss of interaction between two sites that loses CTCF. Are these loci promoters, enhancers etc? Also, I recommend the authors present more individual examples like 2E in supplementary figures.

Response to Reviewer #3, Comment #5:

We thank reviewer for this comment. Together with the response to the Reviewer #3, Comment #8, we changed the example shown in Figure 2E to the *Cebpa* promoter enhancer locus. And we added two more individual examples in new Figure S3B

Figure 2E

Figure legend

(E) UCSC genome browser shot presents HiC interaction frequencies, ChIP seq profiles of ZNF143, CTCF, and H3K27ac, and RNA expression on *Cebpa* locus in both *Znf143 f/f* and *Znf143 -/-* murine HSPCs. The balanced HiC two-dimensional contact matrix is plotted on top. The color intensity presents interaction frequency, while blue circles indicate the *Cebpa* 37kb enhancer promoter loop. ZNF143 and CTCF binding profile are presented below, while H3K27ac histone mark indicates an active chromatin status. The maximum values of Y axis are indicated on the right of each track. Dark triangles indicate ZNF143 and CTCF peaks located on loop anchors. The maximum Y axis value of ChIP-seq signal is set as indicated.

Figure S3B

Figure legend

(B) Snapshots present HiC interaction frequencies, ChIP-seq profiles of ZNF143, CTCF, and H3K27ac, on *Fadd* locus (left panel) and *Nemp1* locus (right panel) in both *Znf143* *ff* and *Znf143* *-/-* murine HSPCs. The balanced HiC two-dimensional contact matrix presents chromatin-chromatin interactions in *Znf143* *ff* (top-right part) and *Znf143* *-/-* (bottom-left part) murine HSPCs. The color intensity presents interaction frequency, while black circles indicate the promoter-enhancer loop. ZNF143 and CTCF binding profile in *Znf143* *ff* or *Znf143* *-/-* murine HSPCs are presented on top, while H3K27ac histone mark indicates an active chromatin status. The maximum values of Y axis are indicated in each track.

~~~~~

**Reviewer #3, Comment #6:**

The authors suggest that ZNF143 regulates CTCF bindings, which in turn regulates chromatin loops. Is there a global assessment regarding how ZNF143-mediated CTCF bindings associate with the ZNF143-mediated chromatin loops?

**Response to Reviewer #3, Comment #6:**

We thank the reviewer for this question. To address this, we performed CTCF motif scanning on ZNF143-mediated loop anchors and ZNF143-unrelated loop anchors, then checked CTCF ChIP-seq signal changes on these regions after loss of ZNF143 loss. In line with higher ZNF143 enrichment detected on ZNF143-mediated loop anchors (Figure 2D), CTCF binding significantly decreased on these regions, indicating that ZNF143 regulates CTCF binding to mediate chromatin loops. We have included this result in new Figure S3A and have described it in the results.

Line 152, Results:

In line with these findings, we also detected significantly decreased CTCF binding on ZNF143-dependent loop anchors, suggesting that ZNF143 directly participated in the maintenance of CTCF-involved promoter-enhancer loops (Figures 2E, S3A).

Figure S3A

Figure S3A legend:

(A) Relative mean plot presents the average changes in the CTCF ChIP-seq signal within  $\pm 2$  kb from CTCF motifs on loop anchors. Loops are divided into different groups as described in Figure 2A. The X axis represents relative distance to the CTCF motifs within loop anchors, while the Y axis represents the  $\log_2$  fold changes of normalized read counts of CTCF ChIP-seq signals in each genomic region. n = two biological replicates.

~~~~~

Reviewer #3, Comment #7:

Figure 2F: the 3C assay seems missing negative controls (regions that doesn't interact with the promoter region of *Cebpa*)

Response to Reviewer #3, Comment #7:

We thank the reviewer for this concern. We have performed new 3C experiment on the *Cebpa* locus with the 76 kb region upstream *Cebpa* promoter as the negative control region and have included the result in the new Figure 2F.

Figure 2F

Figure 2F legend:

(F) The interaction frequencies between the *Cebpa* promoter and the *Cebpa* +37 kb enhancer (“DRE 37 kb”) or with a non-related control region located 18 kb upstream from the *Cebpa* promoter (“URE 18kb”) were examined by 3C qPCR in both *Znf143* f/f and *Znf143* -/- murine HSPCs. The 76 kb upstream region from the *Cebpa* promoter (URE 76 kb) is presented as a negative control region (NC).

~~~~~

**Reviewer #3, Comment #8:**

The *Cebpa* locus should have figures presenting ChIP-seq and HiC tracks (like 2E) that show the chromatin environment.

**Response to Reviewer #3, Comment #8:**

We thank reviewer for this suggestion. Please refer to our response to Comment #5.

~~~~~

Reviewer #3, Comment #9:

It is unclear whether or not a chromatin loop requires CTCF/ZNF143 at both anchors or just one. The model (Figure 5E) presented by the authors show that both anchors need CTCF/ZNF143, but this analysis seems missing in the Figures.

~~~~~

**Response to Reviewer #3, Comment #9:**

We appreciate reviewer for this insight. We agree that current data is not sufficient to prove that ZNF143 is required at both anchors. To match our data more precisely, we have changed our model in new Figure 5E and have pointed this out in the discussion section.

Figure 5E

Line 300, Discussion:

Of note, we do not have a clear answer on whether ZNF143 motifs are required on both anchors, and therefore additional studies with more detailed analysis will be necessary to address this question.

~~~~~

Minor:

Reviewer #3, Minor Comment #1:

It should be noted in the introduction that the previous study from Bailey, 2015 have shown that ZNF143 is essential for formation of chromatin loops at individual loci (3C assays).

Response to Reviewer #3, Minor Comment #1:

We appreciate reviewer for the suggestion. We have included this information in the introduction as a response to the first part of Comment 2 described above.

~~~~~

**Reviewer #3, Minor Comment #2:**

The order of the figures should match the order of the text. Figure 3 is mentioned earlier than Figure 2F-G in the text.

**Response to Reviewer #3, Minor Comment #2:**

We agree and have rearranged the paragraph in the text to match the order of the figures.

~~~~~

Reviewer #4 (Remarks to the Author):

Zhou et al generate a ChIP-grade polyclonal antibody against ZNF143 and ZNF143 f/f mice and perform an elegant series of experiments in ZNF f/f of ZNF f/+ and ZNF -/- mouse HSPCs including ZNF143/CTCF/H3K27Ac ChIP-seq, ZNF143/CTCF re-ChIP, ZNF143/CTCF co-IP, Hi-C, mRNA transcriptomics, E14.5FL flow cytometry and bone marrow transplantation and PB donor chimerism to show that ZNF143/CTCF interactions are common at active promoters/enhancers and loss of ZNF143 results in loss of chromatin loops and gene expression changes within TADs that remain largely unchanged. These changes in mouse HSPCs are associated with changes in the expression levels of key drivers of HSC maintenance/differentiation consistent with the reported phenotypic changes. Overall, the data is clearly presented and convincing.

Comments

~~~~~

**Reviewer #4, Comment #1:**

The focus of the study is mouse HSPCs. The title should reflect this.

**Response to Reviewer #4, Comment #1:**

We thank reviewer for the comment. We have changed the title as described in our response to

Reviewer #1 Comment 7 to:

**ZNF143 mediates CTCF-bound promoter-enhancer loops required for murine hematopoietic stem and progenitor cell function**

**Reviewer #4, Comment #2:**

The numbers of regions bound by ZNF143/CTCF in Fig 1A should be given.

**Response to Reviewer #4, Comment #2:**

We thank reviewer for the suggestion. We have included the number of regions in subgroups in new Figure 1A.

Figure 1A

**Reviewer #4, Comment #3:**

Do regions where CTCF is bound with no ZNF143 binding (Figure 1A) correspond to TADs?

**Response to Reviewer #4, Comment #3:**

We thank reviewer for the question. Please refer to our response to the Comment 1 from Reviewer #3 described above.

**Reviewer #4, Comment #4:**

Any lymphocytes amongst the CD45.2 donor cells in Fig 4?

**Response to Reviewer #4, Comment #4:**

We thank the reviewer for this comment. We have included CD45.2 donor cells derived B cell, T cell, and myeloid cell results in the new Figure S6.

Figure S6

Figure S6

Figure S6 legend

**Supplementary Figure 6. Fetal liver HSPCs fail to reconstitute the hematopoietic system in transplantation after deletion of *Znf143*.**

(A) Dot plot of flow cytometry demonstrating separation of donor-derived cells in peripheral blood from recipients. Cells derived from wild type (*Znf143* f/+) or *Znf143* depleted (*Znf143* -/-) fetal liver cells are indicated as the CD45.2 positive population, whereas competitor derived cells are indicated as the CD45.1 positive population. (B-D) Dot plots describe donor chimerism of B cells (B), T cells (C), and myeloid cells (D) in recipients 16 weeks after transplantation. Each dot represents the percentage of donor chimerism in one recipient mouse. P values in (B-D) are determined by a two tailed unpaired t-test (\*\*\*)  $p < 0.001$ .

~~~~~

Reviewer #4, Comment #5:

CTCF clearly binds DNA without ZNF143 and as the authors data show maintain TADs in its absence in mouse HSPCs- some discussion about this and the absence of change with regards to this higher order chromatin organisation would be welcome.

Response to Reviewer #4, Comment #5:

We appreciate the reviewer for this insight and suggestion. We have included the following paragraph in the discussion to deepen the conversation regarding the maintenance of the TADs and higher-order genome organization.

Line 273, Discussion:

It is quite interesting that ZNF143 mainly regulates CTCF-bound promoter-enhancer loops but not TADs. That loss of ZNF143 only affects CTCF-bound promoter-enhancer loops suggests a novel subgroup of CTCF sites, regulated by ZNF143 and restricted to promoter and enhancer regions. It is widely reported that CTCF and cohesin are the major factors to maintain TAD formation across different cell types^{4-9,15,16,21,26,34,35}. However, in contrast to conservation of TADs in different cell types, loop formations are far more dynamic and regulated not only by CTCF and cohesin, but other factors such as YY1¹¹ and chromatin remodeler Brg1³⁶, among others. Recent studies using high-resolution Micro-C revealed a large

portion of promoter-enhancer loops under 200 bp resolution^{37,38}. Interestingly, these new discovered promoter-enhancer interactions were often detected in the absence of CTCF and cohesin occupancy. Although the biological function remains to be explored, this finding suggests that further studies are required to better understand genome organization and its regulation. Our results in this study strongly suggest that ZNF143 mainly regulates fine-scale genome organization without altering higher-order chromatin structure.

REVIEWERS' COMMENTS

Reviewer #3 (Remarks to the Author):

The authors did a great job addressing the concerns, which has greatly improved the manuscript. I think the manuscript is now ready for publication.